# Optimizing rhBMP-2 Therapy for Bone Regeneration: From Safety Concerns to Biomaterial-Guided Delivery Systems

**DOI:** 10.3390/ijms262110723

**Published:** 2025-11-04

**Authors:** Maria Chernysheva, Evgenii Ruchko, Artem Eremeev

**Affiliations:** 1Koltzov Institute of Developmental Biology of the Russian Academy of Sciences, 119334 Moscow, Russia; chernyshova@idbras.ru (M.C.); ruchko@idbras.ru (E.R.); 2Lopukhin Federal Research and Clinical Center of Physical–Chemical Medicine of Federal Medical Biological Agency, 119435 Moscow, Russia

**Keywords:** rhBMP-2, osteoinduction, bone regeneration, biomaterial carriers, gene therapy, hard tissue reconstruction

## Abstract

Reconstruction of large and complex hard tissue defects remains a major clinical challenge, as conventional autografts and allografts are often limited in availability, biological compatibility, and long-term efficacy, particularly for extensive defects or poor bone quality. Recombinant human bone morphogenetic protein-2 (rhBMP-2) is a potent osteoinductive factor capable of initiating the complete cascade of bone formation. However, its clinical use is restricted by dose-dependent complications such as inflammation, ectopic ossification, and osteolysis. This review synthesizes current evidence on the safety profile of rhBMP-2 and examines strategies to enhance its therapeutic index. Preclinical and clinical data indicate that conventional collagen-based carriers frequently cause rapid burst release and uncontrolled diffusion, aggravating adverse outcomes. It is noteworthy that low doses of rhBMP-2 (0.5–0.7 mg/level in anterior cervical discectomy and fusion (ACDF) or 0.5–1.0 mg/level in transforaminal lumbar interbody fusion (TLIF)) provide the optimal balance of efficacy and safety. Advanced biomaterial-based platforms, such as bioceramic–polymer composites, injectable hydrogels, and 3D-printed scaffolds, enable spatially and temporally controlled release while maintaining osteogenic efficacy. Molecular delivery approaches, including chemically modified messenger RNA (cmRNA) and regional gene therapy, provide transient, site-specific rhBMP-2 expression with reduced dosing and minimal systemic exposure. By integrating mechanistic insights with translational advances, this review outlines a framework for optimizing rhBMP-2-based regenerative protocols, emphasizing their potential role in multidisciplinary strategies for reconstructing complex hard tissue defects.

## 1. Introduction

Spontaneous regeneration of large bone defects resulting from trauma, tumor resections, nonunion fractures, or surgical interventions—including those involving the spine—is exceedingly rare [1]. In such cases, bone autografting has traditionally been employed and remains the clinical standard among the available treatment strategies, as the patient’s own tissue provides inherent osteogenic and regenerative factors. However, even this well-established and effective approach is associated with significant limitations, including the need for additional surgical procedures, invasiveness, postoperative pain, and potential complications at the donor site. Moreover, the applicability of this method to extensive defects is markedly constrained by the limited availability of sufficient graft volume and by age-related declines in bone quality [2]. Consequently, increasing research efforts are being directed toward identifying equally effective but less invasive alternatives. Contemporary strategies for stimulating osteogenesis in orthopedics and neurosurgery include the use of one of the most extensively studied osteoinductive factors—bone morphogenetic protein 2 (BMP-2), a member of the transforming growth factor-β superfamily [3], which was first discovered more than 60 years ago by Marshall Urist [4].

## 2. Biological Functions and Signaling Mechanisms of BMP-2

BMP-2 plays a fundamental role in embryonic development as well as in postnatal tissue repair, particularly in bone regeneration [5]. At the earliest stages of embryogenesis, it orchestrates body patterning by regulating dorsoventral and anteroposterior axis formation, while also guiding somite differentiation and cartilage development of the axial skeleton. Its role extends to central nervous system morphogenesis, where it is essential for both neural tube closure and ocular development, including retinal formation and scleral remodeling [6].

Within the BMP family, which comprises nearly 20 structurally related members, BMP-2 and BMP-7 are recognized as the most potent osteoinductive factors [3]. Among these, BMP-2 exhibits the highest and most reproducible capacity to induce endochondral ossification and bone regeneration. The combination of this pronounced biological activity with its early molecular cloning and successful large-scale recombinant production has made BMP-2 the first BMP to achieve widespread clinical translation. Currently, rhBMP-2 has been evaluated and applied in multiple contexts, including spinal fusion procedures, long bone fracture repair, and maxillofacial augmentation, thereby establishing its role as the benchmark osteoinductive growth factor [3].

Endogenous BMP-2 is synthesized as a precursor protein comprising a signal peptide, a pro-domain, and a mature C-terminal growth factor domain. In the physiological context, the pro-domain facilitates correct folding, dimerization, and secretion, while also regulating latency and extracellular matrix binding. The mature growth factor domain undergoes proteolytic cleavage to release the active dimer, which is sequestered within the bone matrix and becomes bioavailable primarily during bone remodeling or injury [7]. Native BMP-2 therefore acts in a tightly regulated microenvironment, in concert with other matrix-associated molecules such as proteoglycans, growth factors, and matrix metalloproteinases, ensuring spatially restricted and temporally controlled osteogenic signaling [8,9].

RhBMP-2 is produced in heterologous expression systems, most commonly Chinese hamster ovary (CHO) cells or *E. coli*. Recombinant production yields a purified and standardized protein suitable for therapeutic use. Bacteria-derived variants may offer potential advantages in terms of production efficiency and cost. RhBMP-2, produced in CHO cells, is glycosylated—a modification that can affect its stability and interactions. In contrast, *E. coli*-derived rhBMP-2 is non-glycosylated but yields high quantities of the active growth factor domain at lower production costs. Despite this difference, both recombinant variants assemble into disulfide-linked homodimers stabilized by a cystine-knot motif [10]. Thus, both CHO- and *E. coli*-derived rhBMP-2 can form biologically active homodimers, but the two variants differ in their post-translational modifications and glycosylation, which can influence protein folding, stability, and pharmacokinetics. Clinical investigations indicate that both forms of rhBMP-2 demonstrate osteoinductive efficacy [3,11].

Both native and recombinant BMP-2 variants signal through heteromeric complexes of type I and type II BMP receptors, activating both Smad-dependent and Smad-independent signaling cascades. In the canonical pathway, receptor-mediated phosphorylation of Smad1/5/8 allows complex formation with Smad4 and subsequent nuclear translocation, where transcription of key osteogenic factors such as runt-related transcription factor 2 (RUNX2) and Osterix is induced [12]. In certain contexts, such as embryonic or tumor cells, BMP-2 can also engage Smad2/3, underscoring its signaling plasticity. Parallel Smad-independent pathways, including (mitogen-activated protein kinase) MAPK and PI3K/AKT, further regulate cell proliferation, survival, and apoptosis [13], contributing to osteogenesis but also posing risks under conditions of excessive stimulation [11]. The activity of BMP-2 is tightly regulated at multiple levels. Extracellular antagonists, including noggin, sclerostin, follistatin, and chordin, sequester the ligand to prevent receptor binding, while extracellular matrix components such as fibronectin and tenascin-C modulate BMP-2 bioavailability. Intracellularly, inhibitory Smad6 and Smad7 restrain downstream signaling [14]. Moreover, BMP-2 interacts with other growth factors in context-dependent ways. For instance, co-administration with platelet-derived growth factor (PDGF) attenuates the osteogenic response [15]. Given this multifaceted regulation and potent biological activity, it is unsurprising that BMP-2 has become a focal point in regenerative medicine research. Not only can it orchestrate the recruitment and differentiation of mesenchymal cells into osteoblasts [12], but it also drives the entire cascade of endochondral ossification [16]. These insights have laid the foundation for the development of rhBMP-2, which remains the only osteoinductive agent (protein) approved by the U.S. Food and Drug Administration (FDA) for use in orthopedic and spinal surgery since its initial approval in 2002 [17,18]. Its clinical utility has been particularly evident in spinal fusion and craniofacial surgery, where it accelerates healing, reduces the need for reoperation, and shortens the recovery time [19,20,21]. The major molecular mechanisms by which BMP-2 regulates osteogenic differentiation are illustrated in Figure 1, highlighting both canonical SMAD-dependent signaling and non-canonical pathways (MAPK, PI3K/Akt), as well as their interaction with extracellular antagonists and the related TGF-β–SMAD2/3 cascade.

## 3. Expression Systems and Technological Approaches for the Production of rhBMP-2

Functionally, rhBMP-2 differs from native BMP-2 in its mode of presentation. Physiologically, BMP-2 is stored within the extracellular matrix of bone in a latent form. It is present at low concentrations and acts synergistically with other matrix-bound signaling molecules. Unlike endogenous BMP-2, which is locally sequestered and released in response to physiological cues, therapeutic rhBMP-2 is typically delivered at supraphysiological concentrations on a biomaterial carrier. This distinction underlies its powerful osteoinductive effect but can also lead to dose-dependent adverse outcomes [10,22].

Building on the clinical success of rhBMP-2, attention has turned to optimizing its production. Large-scale application requires manufacturing platforms that are scalable, reproducible, and economically feasible. The choice of expression system critically affects not only protein yield but also folding, post-translational modifications, and ultimately biological activity.

A number of comparative studies illustrate these challenges. For instance, Jérôme et al. [18] examined stable CHO cells, transiently transfected human embryonic kidney 293 (HEK293) cells, and a CHO lysate-based cell-free system. While CHO cells provided high-quality protein, the yields were extremely low (≈153 pg/mL). In contrast, HEK293 cells produced much higher amounts (≈280 ng/mL) while maintaining activity, albeit with transient expression. The cell-free CHO system proved most efficient, generating ≈40 µg/mL in just 3 h and allowing precise control over protein modifications [23]. A different strategy, repeated transient transfection (RTT), further improved productivity. Riedl et al. showed that RTT in HEK293 suspension cells enabled sustained rhBMP-2 expression for up to 360 h and titers of 509 ng/10^6^ cells—far surpassing the yields from stable CHO lines [24]. Beyond mammalian systems, plant-based approaches have also been explored. Nguyen et al. successfully engineered rice cells to stably produce rhBMP-2 through clustered regularly interspaced short palindromic repeats (CRISPR)/Cas9-mediated knock-in of a codon-optimized construct, achieving up to 21.5 µg/mL with native post-translational modifications and sustained activity across generations [25].

Parallel to improvements in expression platforms, protein engineering strategies aim to enhance solubility, receptor selectivity, and stability. For instance, the hydrophilic variant of rhBMP-2 additionally containing an improved heparin binding site was designed to increase hydrophilicity and heparin affinity, allowing high-yield soluble expression in *E. coli* SHuffle strains without complex refolding steps [26]. Similarly, Chen et al. developed a periplasmic *E. coli* system using thioredoxin fusion and alkaline phosphatase signal peptides, yielding 6.2 mg/L of bioactive rhBMP-2 [27]. Recently, Kang et al. engineered cell-permeable rhBMP-2 with an artificial macromolecular transduction domain, achieving superior bone regeneration in murine, rabbit, and equine models without ectopic ossification [28]. Together, these studies underscore that advances in expression technology and protein engineering are not merely technical refinements: they directly shape the translational potential and clinical accessibility of rhBMP-2.

### Integration of rhBMP-2 into Modern Therapeutic Protocols

With improved production methods, clinical research has focused on how rhBMP-2 can be most effectively integrated into therapeutic strategies. Despite being approved for clinical use more than two decades ago, rhBMP-2 continues to attract attention. Ongoing studies refine its indications, delivery methods, and safety profile.

In spinal surgery, randomized clinical trials (RCTs) have consistently highlighted its potential. For example, in patients with multisegmental spinal deformities, rhBMP-2 combined with hydroxyapatite (HA) achieved 100% fusion at both 6 and 12 months, with parallel improvements in Visual Analogue Scale, SF-36, and SRS-22 scores compared with HA alone [29]. Dental applications have shown similarly compelling results. An rhBMP-2 delivered on a biphasic calcium phosphate (BCP) scaffold induced nearly 94% new bone formation at 9 months in an RCT on alveolar ridge preservation. This result is substantially higher than the 23% observed in the control group [30]. Wei et al. further demonstrated that rhBMP-2 delivered via a β-tricalcium phosphate (β-TCP)-based BioCaP coating provided sufficient bone density for stable implant placement within just 6 weeks without systemic side effects [31]. These findings align with the evidence from large-scale syntheses. A meta-analysis of 14 RCTs (789 rhBMP-2 patients vs. 727 iliac crest bone graft (ICBG) patients) confirmed that rhBMP-2 improved fusion rates in posterolateral spinal fusion, shortened operative time, reduced blood loss, and lowered the risk of donor-site complications. The authors highlighted the particular value of rhBMP-2 in situations where autografting is technically or clinically challenging [32].

## 4. Risks and Adverse Effects of rhBMP-2 Therapy

The clinical use of rhBMP-2 has been consistently associated with a distinct profile of adverse events, raising considerable concern within the medical community [33]. This concern has fueled ongoing research, with nearly 100 clinical trials registered specifically for the pharmaceutical form of rhBMP-2, dibotermin alfa. The findings from these studies have delineated a range of complications that limit a broader application of dibotermin alfa: these can be categorized as acute inflammatory responses, long-term issues related to disordered bone formation, and debated systemic risks.

Among the most common acute complications are local inflammation and soft tissue swelling. Long-term adverse outcomes, by contrast, are frequently related to disordered osteogenesis, including ectopic or heterotopic ossification, osteolysis, and adipogenesis leading to abnormal fat deposition at the implantation site [34].

Imaging studies consistently identify heterotopic ossification as the most frequent finding after lumbar fusion with rhBMP-2 [35]. Other adverse events, generally affecting a minority of patients, include soft-tissue swelling and seroma/hematoma, chemical radiculitis/nerve-root irritation, and early osteolysis or endplate cysts/subsidence, particularly when supratherapeutic protein loads or extra-cage diffusion occur. Mechanistically, burst release of the protein can stimulate osteoclast-mediated resorption and peri-implant cyst formation; judicious dosing and containment mitigate this risk [36].

Safety concerns are magnified in specific anatomical contexts. In the cervical spine, for instance, the FDA has issued a warning about life-threatening airway edema with off-label use, and recent summaries continue to recommend conservative dosing and strict localization in this region [17].

The potential role of rhBMP-2 in oncogenesis remains a particularly nuanced and controversial area [37]. The evidence presents a duality: while growing data suggest that rhBMP-2 may promote tumor progression [38]—for example, by stimulating the epithelial–mesenchymal transition in breast cancer [39] or by activating cancer stem cells in glioma [40]—other reports indicate context-specific tumor-suppressive effects, as observed in colorectal cancer [41]. This complexity is reflected in clinical data. An earlier high-dose (40 mg) lumbar arthrodesis trial reported a higher incidence of new cancers versus control. In contrast, more recent thoracolumbar cohorts and registry-scale analyses at standard clinical doses generally do not demonstrate an increased malignancy risk. This suggests that any theoretical oncogenic signal is likely dose-contingent rather than inherent to the rhBMP-2 molecule itself [42].

Overall, contemporary reviews confirm that while rhBMP-2 (e.g., Infuse) can enhance fusion and early bone formation, its characteristic complication profile tightens the therapeutic window. Consequently, the incremental gains in fusion probability must be balanced against these quantitatively described risks. This risk–benefit calculus favors the adoption of low-dose protocols, controlled-release carriers, and meticulous containment strategies tailored to the anatomical corridor and patient-specific risk factors, such as bone quality, metabolic status, and vascularity.

Taken together, these observations underscore the complexity of rhBMP-2 biology and highlight the critical need for a careful monitoring of both acute and delayed outcomes. While rhBMP-2 remains a powerful osteoinductive agent, its therapeutic application demands a cautious approach. Continuous surveillance, refinement of clinical indications, and prudent patient selection are paramount to ensuring that its significant benefits reliably outweigh its potential harms. The contrasting approaches of regional gene therapy and conventional rhBMP-2 delivery in bone regeneration are illustrated in Figure 2.

### 4.1. Local Inflammation and Edema

One of the most significant limitations to the widespread clinical use of rhBMP-2 remains its pro-inflammatory activity, which can provoke local inflammatory responses as well as systemic complications. This issue becomes particularly relevant in contexts beyond orthopedic surgery, where the biological activity of rhBMP-2 manifests in tissues with distinct immune and reparative environments [43]. A notable example comes from regenerative cardiology, where interest in BMP-2 arises from its potential to modulate mesenchymal cell differentiation and stimulate myocardial repair. Wang et al. demonstrated that pre-incubation of c-Kit^+^ mesenchymal stem cells with BMP-2 in vitro promoted their differentiation into cardiomyocyte-like cells, which subsequently improved contractile function following transplantation into a rat model of myocardial infarction [44]. Building on this approach, Guo et al. developed a functionalized self-assembling peptide incorporating motifs for cell adhesion and BMP-2 binding. When combined with c-Kit^+^ mesenchymal stem cells, this construct enhanced myocardial regeneration, angiogenesis, and cardiac function, and also reduced fibrosis in a rat infarction model [45].

By contrast, direct administration of BMP-2 into the ischemic myocardium has yielded less encouraging results. Pulkkinen et al. investigated intramyocardial injection of an adenoviral vector encoding human BMP-2 in a rat model of chronic myocardial ischemia. Although gene expression was confirmed in the myocardium, no improvements in angiogenesis, cardiomyocyte proliferation, or contractile function were observed. Moreover, the animals receiving an adenoviral vector containing the BMP2 gene exhibited signs of an enhanced inflammatory response, including pericardial effusion and infiltration of CD3^+^ T lymphocytes at the injection site. While no systemic toxicity or histopathological abnormalities were detected in other organs, the pronounced local immune reactions highlighted a potentially unfavorable safety profile for direct BMP-2 application during chronic ischemic heart disease [46].

Similar pro-inflammatory effects of rhBMP-2 have also been documented in surgical practice, particularly in anatomically sensitive regions such as the cervical spine. Preclinical data confirm that rhBMP-2 administration can induce pronounced local inflammatory and morphological changes, including hematoma, seroma, soft tissue infiltration, and edema. Clinically, these effects may compromise vital structures, leading to complications such as dysphagia, dysphonia, and, in severe cases, airway stenosis. Such outcomes have been repeatedly described in large clinical cohorts and systematic reviews [47,48], prompting the FDA in 2008 to issue an official safety warning regarding the risks of rhBMP-2 use in cervical spinal fusion [49].

### 4.2. Hyperostosis and Osteolysis

Therapy with rhBMP-2 is associated with several serious long-term adverse effects related to disordered osteogenesis. Numerous clinical studies have reported a considerable risk of hyperostosis and heterotopic bone formation [37]. Computed tomography and magnetic resonance imaging series note ectopic bone in a substantial fraction of cases, in some reports up to ~75%, although most of these cases are radiographic and asymptomatic [35]. These radiographic findings typically become apparent within the first 3–6 months postoperatively [42,50,51]. These ectopic bone deposits are typically linked to protein diffusion beyond the implantation site. The structure of the newly formed tissue often displays inferior quality.

One possible explanation lies in the dual role of rhBMP-2, which can drive not only osteogenic but also adipogenic differentiation of mesenchymal stem cells. This process results in excessive lipid accumulation and the development of bone cysts, where bone volume may appear to increase but the trabecular architecture is compromised: trabecular spacing is reduced and overall bone strength declines [8]. Particularly concerning is the occurrence of hyperostosis in the spine, where abnormal bone overgrowth can compress neural structures and necessitate revision surgery [52]. This clinically significant hyperostosis is often a late complication, manifesting 6–12 months after surgery, and its risk correlates with higher doses (e.g., >2.0 mg per level in the cervical spine) and the use of certain collagen-based carriers that allow for uncontrolled diffusion [50,53]. Pathogenetically, this effect is linked to excessive activation of SMAD-dependent signaling pathways. Notably, the risk of hyperostosis persists even with the use of modern carriers designed for controlled release [54].

Equally troubling is the phenomenon of osteolysis, especially in spinal surgery. BMP-2-induced activation of osteoclasts may lead to trabecular bone resorption, graft subsidence, and loss of implant stability [14]. Clinical studies report osteolytic events in 20% to 70% of cases where rhBMP-2 is applied. Osteolysis is predominantly an early complication, peaking at 2–6 weeks postoperatively and often stabilizing by 3–6 months [53,55]. Patients frequently develop pronounced osteolytic defects and cysts at the implantation site, most commonly in the early postoperative period [42]. The severity of osteolysis is strongly dose-dependent, with higher doses being major risk factors [55,56]. Taken together, these complications are strongly associated with excessive dosing and uncontrolled protein diffusion within tissues, underscoring the urgent need for delivery systems that ensure a spatially and temporally regulated release of rhBMP-2 in clinical practice.

## 5. New Clinical Applications of rhBMP-2

The scope of clinical applications for rhBMP-2 has expanded well beyond its traditional use in orthopedics and maxillofacial surgery. Increasing attention is now directed toward its integration into more complex reconstructive procedures, particularly in spine surgery, where severe deformities often require three-column osteotomies or revision operations. Interestingly, this broader application has been accompanied by a trend toward dose reduction, reflecting efforts to minimize the risk of adverse effects while preserving therapeutic efficacy. For instance, Bannwarth et al. reported that the mean dose per patient of rhBMP-2 for adult spinal deformity (ASD) surgery decreased from 26.6 to 20.7 mg (*p* < 0.001), which was associated with nearly a twofold reduction in complication rates requiring reoperation. Importantly, nonunion rates remained stable, suggesting that a more moderate and balanced approach to rhBMP-2 use may be appropriate in challenging surgical scenarios [57]. Nevertheless, not all studies have demonstrated the clear superiority of rhBMP-2 over alternative biologics. Onafowokan et al. found that in patients with spinal deformities, rhBMP-2 did not provide statistically significant advantages in terms of clinical outcomes or quality of life when compared with more accessible options, such as a combination of bone marrow aspirate, cancellous bone, and the synthetic osteoactive matrix i-Factor. Moreover, treatment with rhBMP-2 was associated with higher costs and a potentially increased risk of neurological complications, underscoring the need for a further optimization of dosing strategies, carrier selection, and personalized treatment protocols [58].

Beyond dosage considerations, innovations in delivery systems are also shaping the future of rhBMP-2 applications. For example, Lin et al. recently introduced a biodegradable intramedullary implant designed for sustained rhBMP-2 release. In a rat bone transport model, this construct accelerated osteogenesis and defect consolidation, promoted strong osseointegration, and reduced complication rates such as infections and nonunions, highlighting its promise for enhancing both the efficacy and safety of distraction osteogenesis [59]. Another promising direction is the integration of rhBMP-2 into minimally invasive surgical approaches. Chen et al. evaluated the use of rhBMP-2 adsorbed onto an HA carrier in posterior extreme lateral interbody fusion (PE-PLIF), a modified minimally invasive technique. The rhBMP-2/HA combination not only achieved high fusion rates but also reduced interbody implant migration and vertebral body displacement, leading to a greater construct stability. Patients in the rhBMP-2 group experienced significant functional improvements without evidence of rhBMP-2–related complications [60].

In parallel, efforts are underway to identify cost-effective and potentially safer osteoactive substitutes that could complement or even replace rhBMP-2 in selected contexts. One such development is OSTEOAMP, an allogeneic graft enriched with endogenous growth factors designed to stimulate fusion. In a large retrospective study of 1154 patients undergoing single- or multi-level PLIF or TLIF for degenerative lumbar disease, Hani et al. compared the effectiveness and safety of rhBMP-2 and OSTEOAMP. Over a 24-month follow-up, no significant differences were observed in overall fusion rates or the incidence of pseudarthrosis requiring revision surgery. Both groups also demonstrated comparable functional improvements according to the Oswestry Disability Index (ODI), Numeric Rating Scale (NRS) for pain, and EQ-5D scores at 12 months, suggesting that OSTEOAMP may represent an effective and clinically safe alternative to rhBMP-2 in lumbar fusion procedures [61].

### Translational Barriers: Regulatory Approval, Cost, and Scalability

Despite promising efficacy, rhBMP-2 therapies face significant translational hurdles. Regulatory agencies treat rhBMP-2 products as complex biologic–device combinations, requiring extensive safety and efficacy trials for each new formulation. Infuse Bone Graft, for instance, is FDA-approved only for specific indications, and any novel rhBMP-2 delivery system must undergo similarly rigorous approval. A realistic appraisal of translational barriers should note that rhBMP-2 approvals are indication- and device-specific, not blanket. In the United States, the FDA authorized InFUSE Bone Graft only in combination with the LT-CAGE for single-level ALIF, with subsequent supplements tied to specific implants rather than broad anatomic expansion; off-label use in the cervical spine was explicitly flagged in the FDA’s 2008 public health communication because of life-threatening airway edema, reinforcing the need for conservative dosing and strict localization in that region [18]. In the EU, InductOs (dibotermin alfa) likewise holds indication-specific marketing authorization, and its vulnerability to GMP disruptions was illustrated when the European Medicines Agency recommended suspension in 2015 due to quality defects at a component site; the European Commission lifted the suspension in 2017 only after GMP compliance was restored, showing how manufacturing robustness directly conditions clinical availability [62].

Economic factors further constrain adoption. rhBMP-2 remains very costly: Infuse kits still run on the order of $2500–$6000 each [63], a price that has not decreased in over 20 years. A high acquisition cost often outweighs the perceived savings; some analyses report that autograft procedures (despite longer OR times) can match or beat total cost once all follow-up care is counted. For example, one study found that when revision surgeries were included, the net costs of rhBMP-2 versus autograft were roughly equivalent. Large-scale manufacturing of rhBMP-2 under GMP also remains complex and expensive, limiting the production scale [64]. A 2024 systematic review of cost-effectiveness analyses reported a baseline hospital acquisition cost for rhBMP-2 ranging from approximately $900 to $5500 per case, varying by indication, dose, and institutional contracting. The review further noted that rhBMP-2 frequently entails higher upfront costs compared with autograft or demineralized bone matrix with local bone, for example $42,627 vs. $38,686 at 1 year in a large lumbar fusion cohort and $97,917 vs. $85,838 at 90 days in multilevel cervical fusion. These findings underscore that although rhBMP-2 can reduce revision surgery rates in selected high-risk patients, its routine use may impose a significant budget impact and is not uniformly cost-effective across all clinical scenarios [65]. Taken together, these data rationalize the common practice of reserving rhBMP-2 for scenarios in which a marginal gain in fusion probability materially alters outcomes—multilevel constructs, revisions, limited autograft, or compromised bone quality. Scalability hinges on complex, tightly regulated biomanufacturing. Sharp differences in production yields across expression platforms have been documented (e.g., stable CHO ≈ 153 pg/mL vs. HEK293 ≈ 280 ng/mL vs. CHO cell-free ≈ 40 µg/mL in ~3 h), illustrating why a consistent GMP supply remains costly and why combination-product kits remain expensive in practice; the EU suspension episode above underscores the real-world consequences of supply-chain fragility.

Given these considerations, rhBMP-2 use must be tailored to patient and clinical context. In spine fusion and fracture repair, rhBMP-2 typically offers only a modest absolute improvement in fusion rates over autograft. In routine cases (young, healthy patients or small defects), standard grafts are often preferred to avoid rhBMP-2’s costs and risks. Within rhBMP-2 use, the advent of new carriers allows dose and delivery to be customized. For example, a slow-release ceramic or hydrogel scaffold can be chosen for an elderly patient to enable a low BMP-2 dose with a sustained effect, whereas a more aggressive rhBMP-2 regimen might be used for a large defect in a younger patient. Clinicians should apply quantitative data to decision-making: knowing that a keratin matrix retains more rhBMP-2 locally [66] or that a putty carrier doubles fusion success in models can guide carrier choice. In all cases, the goal is to match the rhBMP-2 platform to the defect and to patient factors, leveraging superior carriers and dosing protocols to maximize bone healing while minimizing ectopic bone and inflammation [67].

## 6. Patient-Oriented Strategies for rhBMP-2 Application

In the era of personalized medicine, there is a growing need for more precise patient selection and preparation in order to identify those who may benefit most from rhBMP-2 therapy. This has stimulated interest in studies aimed at elucidating the determinants of tissue responsiveness, including the local immune status, the microenvironment, and patient-specific healing characteristics. The importance of tailoring therapy to the individual is underscored by the meta-analysis conducted by Laurie et al., which included data from 10 industry-sponsored RCTs. The authors demonstrated that the effectiveness of rhBMP-2 in lumbar fusion was strongly influenced by clinical and demographic profiles: the most pronounced benefits were observed in smokers, patients under the age of 60, and individuals with a normal body mass index (*p* < 0.01). By contrast, patients who were older, obese, or non-smokers did not experience the same therapeutic advantage [68]. Additional insights into the somatic predictors of outcomes were provided by Overley et al., who found that hypertension was the only significant risk factor for radiographic pseudarthrosis following minimally invasive TLIF with either rhBMP-2 or cellular bone matrix. Interestingly, factors commonly assumed to impair healing, such as smoking, obesity, and diabetes, showed no statistically significant association with nonunion risk, highlighting the importance of vascular health for successful osteoinduction [69].

Bone quality itself has also emerged as a critical stratification parameter. In a study by Kim et al., patients with low bone mineral density who underwent TLIF with rhBMP-2 achieved accelerated fusion formation compared with controls, particularly within the first six months postoperatively. These findings emphasize the value of rhBMP-2 in osteopenic conditions, where impaired bone quality might otherwise compromise healing [70]. Expanding beyond individual risk factors, rhBMP-2 has also gained traction in the management of complex spinal deformities. Multilevel fusions, which are often necessary for the surgical correction of adult sagittal imbalance (ASD), are known to carry an elevated risk of pseudarthrosis. In this context, lateral lumbar interbody fusion (LLIF) augmented with rhBMP-2 has been proposed as a safe and effective strategy. Singh et al. reported fusion success in 94.5% of patients (169 of 179; mean age 65.3 years), with only 1.1% requiring revision surgery for symptomatic pseudarthrosis [71].

Yet, despite these advances and the ongoing refinement of clinical protocols, both researchers and clinicians are increasingly attentive to safety concerns. Accumulated clinical experience has revealed a spectrum of adverse events associated with rhBMP-2 use, emphasizing the need for careful risk–benefit evaluation, patient stratification, and ongoing surveillance in order to ensure optimal and safe outcomes. An overview of the approved and investigational clinical applications of rhBMP-2 is presented in Figure 3.

The optimal rhBMP-2 dose and delivery system should not be considered in terms of fixed parameters but rather tailored to the surgical indication and the biological profile of the patient. In long-bone trauma, pivotal randomized trials have shown that in open tibial fractures, a total dose of 6–12 mg (0.75–1.5 mg/mL) applied via an absorbable collagen sponge accelerates fracture union and reduces the need for secondary interventions, compared with standard autograft-based care [78]. In the field of oral and maxillofacial surgery, maxillary sinus floor augmentation has consistently demonstrated reliable bone formation with doses of 0.75–1.5 mg/mL, while post-extraction socket preservation benefits from slightly higher concentrations, which translate into greater ridge volume and improved implant stability [79]. For spinal fusion procedures, rhBMP-2 is typically applied at 1.0–1.5 mg/mL per fusion level; although its use remains off-label in several jurisdictions, clinical evidence supports improved fusion rates and reduced reoperation risk, particularly in multilevel constructs and adult deformity correction [80].

Beyond the surgical indication itself, patient-specific variables play a critical role in determining the most appropriate rhBMP-2 regimen. Factors such as age, metabolic status, bone quality, smoking, and comorbidities (including diabetes or osteoporosis) directly influence osteoinductive responsiveness. For example, experimental and clinical data suggest that higher doses of rhBMP-2 may partially compensate for impaired bone regeneration in a segmental femoral defect model in diabetes mellitus BB Wistar rats [81], while in osteoporotic conditions, slow-release carriers (e.g., ceramic composites, hydrogel matrices, or hybrid scaffolds) help maintain local bioactivity, reduce systemic exposure, and improve graft stability [82]). Moreover, vascular health has emerged as an important stratification parameter, as impaired perfusion limits rhBMP-2-mediated healing despite adequate dosing [83]. These insights highlight the fact that rhBMP-2 therapy is most effective when embedded into a precision medicine framework, where dosage and delivery vehicles are selected not only according to the anatomical site but also according to the patient’s biological and clinical profile. Such individualized strategies maximize efficacy while reducing complications such as ectopic bone formation, infection, or inflammatory reactions, thereby reinforcing the translational relevance of rhBMP-2 in modern regenerative medicine. Such considerations provide the conceptual basis for tailoring rhBMP-2 therapy in clinical practice. Table 1 summarizes representative studies across diverse surgical indications, highlighting typical dose ranges, delivery vehicles, patient-specific variables, and key outcomes. This comparative overview underscores the fact that rhBMP-2 efficacy is context-dependent and illustrates how adjusting dosage and scaffold selection according to the clinical scenario and patient characteristics can optimize bone regeneration while minimizing complications.

**Table 1 ijms-26-10723-t001:** Summary of clinical studies using rhBMP-2 for bone regeneration across a range of indications.

Indication	rhBMP-2 Dose	Delivery System	Patient Factors	Key Outcomes	Citation
Lumbar spinal fusion (adult spinal deformity, 3–5 levels)	~3.0 mg per level (≈9–15 mg total)	Hydroxyapatite (HA) carrier + rhBMP-2 (no autograft)	Adults (19–80 y) with multilevel deformity (L1–S1); osteoporosis excluded	100% fusion at 6 and 12 mo with rhBMP-2 + HA vs. 88% with HA alone; significant pain/QoL improvement (no adverse events).	[84]
Lumbar interbody fusion (degenerative TLIF, 1–2 levels)	0.5–1.0 mg per segment	rhBMP-2 (*E. coli* -derived) + hydroxyapatite/bone graft in interbody cage	Adults with degenerative lumbar disease undergoing 1–2 level TLIF	100% fusion at 52 and 104 weeks; significant improvement in ODI and VAS; no rhBMP-2-related complications.	[73]
Maxillary sinus augmentation (for dental implants)	~1.5 mg/mL (≈6–12 mg per sinus graft)	rhBMP-2 with HA scaffold or ACS and bone graft	Edentulous posterior maxilla with atrophic ridge (low residual bone height)	90–100% implant survival with rhBMP-2 vs. 86–95% without; comparable graft height gain; rhBMP-2 group showed less marginal bone loss.	[85,86]
Alveolar ridge (socket) preservation (tooth extraction)	~0.3 mg per socket (1.5 mg/mL, 0.2 mL)	Absorbable collagen sponge (ACS) soaked with rhBMP-2	Adults undergoing posterior tooth extraction	Significantly less buccal bone resorption (1.8 mm less height loss vs. control) at 12 weeks; better bone fill/volume retention; no complications reported.	[87]
Long-bone fracture nonunion (femur, tibia, humerus)	≤6 mg per site (avg ~ 5 mg)	HA granules + autologous cancellous bone, all mixed with rhBMP-2	Adults with atrophic/oligotrophic or infected nonunions	95.8% union at 6 mo, 100% at 12 mo; marked improvement in function and pain; no adverse events or antibodies.	[84]
Alveolar cleft reconstruction (cleft lip/palate patients)	rhBMP-2 with demineralized bone matrix (DBM) (dose varied)	DBM carrier + rhBMP-2 in cleft site	Children with unilateral/bilateral alveolar cleft	Identified “critical-size” cleft volume: graft failure rises sharply above ~885 mm^3^. Below that volume, rhBMP-2/DBM success ≈ 86%; above it, failure ≈ 65%.	[88]

## 7. Carriers as Critical Determinants of rhBMP-2 Efficacy and Safety

A critical determinant of the therapeutic profile of rhBMP-2 is the choice of carrier, which must provide both controlled release and spatial stabilization of the protein. A wide range of carriers has been evaluated in experimental and clinical settings, ranging from resorbable collagen matrices to synthetic bioceramic scaffolds. Among natural biomaterials, collagen is regarded as the “gold standard” for rhBMP-2 immobilization, and collagen-based matrices currently dominate clinical practice. For instance, the collagen sponge-based INFUSE™ Bone Graft has been clinically approved for the treatment of acute fractures and for spinal fusions.

Nevertheless, the use of collagen carriers is accompanied by significant limitations. To achieve therapeutic efficacy, rhBMP-2 must be administered at doses far exceeding physiological levels—milligrams rather than micrograms of protein. Moreover, the rapid release of rhBMP-2 from such scaffolds produces a “burst” of local cytokine concentration, which can trigger inflammation, ectopic bone formation, and disturbances in physiological tissue remodeling [89]. Experimental data provide further support for these concerns: in a rat femoral defect model, low concentrations of rhBMP-2 (5–10 µg/mL) failed to induce fusion, intermediate doses (~30 µg/mL) promoted normal regeneration, whereas high doses (150–600 µg/mL) resulted in cyst-like lesions, soft tissue edema, and inflammatory infiltrates [90]. Additional risks arise from systemic distribution. In a murine model, local rhBMP-2 application was shown to enter the bloodstream, leading to the development of hepatic tumors due to the proliferation of myelodysplastic syndrome (MDS) cells and increased secretion of IL-6 [38].

At the molecular level, excessive concentrations of rhBMP-2 have been reported to suppress the expression of genes critical for osteogenesis, chondrogenesis, cell adhesion, and cell migration, while simultaneously activating pro-inflammatory pathways. For example, overexpression of integrin subunit beta, a gene associated with leukocyte chemoattraction, has been linked to the formation of cystic cavities and hematomas. Furthermore, altered expression of annexin V and cxcl10 has been associated with the inhibition of mineralization and the stimulation of osteoclastogenesis, respectively. These findings highlight the necessity of strict control over both dosing and spatial localization of rhBMP-2. The design of delivery systems capable of achieving temporally and spatially regulated release remains central to balancing therapeutic efficacy with safety in clinical practice [77].

Recent preclinical and clinical data indicate that rationally engineered carriers can equal or surpass the performance of absorbable collagen sponges while attenuating adverse effects by localizing and smoothing rhBMP-2 release. Poly(ethylene glycol) “click” hydrogels achieved defect closure equivalent to collagen in murine calvaria yet produced less off-target mineralization, while tunable TG-PEG hydrogels similarly enhanced calvarial regeneration with stiffness-dependent delivery kinetics [91]. Keratin-based scaffolds have matched collagen for new-bone formation in a rat femoral defect while exhibiting ~4-fold higher local retention of fluorescently labeled rhBMP-2 and reduced distal biodistribution, consistent with improved containment [66]. Composite biphasic-ceramic/hydrogel putties that sustain rhBMP-2 release (e.g., hydroxyapatite/β-TCP/poloxamer formulations) have outperformed collagen sponges in a rat femoral nonunion model, yielding a 6-week union rate of 76.5% vs. 35.3% and higher BV/TV and BMD [67]. Early clinical experience aligns with this trajectory: in a 2024 prospective multicenter TLIF study using *E. coli*-derived rhBMP-2 (0.5–1.0 mg per level with an HA putty), surgeons reported 100% CT-confirmed fusion at 52 and 104 weeks with no cases of seroma, radiculitis, cage migration, or ectopic bone [73]. Together, these findings support the premise that controlled-release matrices can maintain or enhance osteogenesis at a given rhBMP-2 dose, thereby reducing total protein burden and the likelihood of inflammatory or heterotopic responses, while broadening the therapeutic window.

### 7.1. Optimization of rhBMP-2 Dose and Delivery

Considerable attention has been directed toward optimizing dosing regimens and delivery strategies, particularly the development of localized and controlled release systems. A central focus has been on reducing the therapeutic dose in order to lower the frequency of adverse effects while preserving osteoinductive efficacy. For example, a large RCT from China involving more than 1100 patients compared low-dose rhBMP-2 (0.5 mg/level) with the traditional ICBG in ACDF. At three months, the rhBMP-2 group showed earlier fusion, although by 6 and 12 months the outcomes were comparable, suggesting that low-dose rhBMP-2 may represent a viable alternative to autografting in ACDF. Importantly, complication rates such as dysphagia were similar across the groups, and no severe adverse events, including oncogenic transformation and infection, were observed [74]. Further evidence comes from an RCT of 74 patients undergoing single-level posterolateral lumbar fusion (PLF), in which rhBMP-2 expressed in *E. coli* and immobilized on an HA carrier achieved higher fusion quality and a markedly lower rate of nonunion compared with ICBG [72]. Consistent with these findings, Kwon and colleagues evaluated controlled dosing in one- and two-level TLIF, applying 0.5–1.0 mg/level of rhBMP-2 with an HA carrier. In 30 patients with degenerative spinal disease, this approach yielded a fusion rate of ≈98% without major complications, supporting the value of moderate dosing in combination with localized delivery systems [73]. Complementary insights are provided by retrospective data. Mendenhall et al. analyzed outcomes in 198 patients who underwent ACDF with low-dose rhBMP-2 (mean 0.5 mg/level). Their results demonstrated high effectiveness: 96% complete arthrodesis within 15 months, together with acceptable complication rates (dysphagia 11%, neck swelling 6%, and pseudarthrosis 1%) [75].

Finally, broader trends have been captured in large-scale reviews. A systematic review by Wen et al., encompassing 29 studies with more than 1.5 million patients, indicated that doses ≤0.7 mg/level offer the most favorable balance between safety and efficacy, whereas higher doses were increasingly associated with complications such as dysphagia, surgical site infection, and impaired wound healing, with only doses exceeding 2.1 mg/level being consistently linked to elevated rates of early postoperative infection in cervical spine surgery [76]. Table 2 provides a summary of strategies for optimizing rhBMP-2 dosing in clinical studies.

**Table 2 ijms-26-10723-t002:** Optimization of rhBMP-2 dosing in clinical studies.

Procedure	Dose of rhBMP-2 (mg/level)	Study Design	Efficacy (Fusion/Timeline)	Complications	Reference
ACDF	0.5	RCT, >1100 patients	Faster fusion by 3 months; differences with autograft diminished by 6–12 months	Similar dysphagia rates in both groups; no serious adverse events reported	[74]
ACDF	~0.5 (average dose)	Retrospective, *n* = 198	Complete arthrodesis in 96% (≈15 months)	Dysphagia 11%, neck swelling 6%, pseudarthrosis 1%	[75]
ACDF	≤0.7 optimal; ≥0.7: increased risk	Systematic review/ meta-analysis (29 studies; 1,539,021 patients)	Best balance of efficacy/safety at ≤0.7 mg/level	At levels ≥ 0.7 mg/L: higher risk of dysphagia, infections, and wound healing complications	[76]
Cervical spine (posterior approach)	>2.1 associated with a higher risk of infections	Meta-analysis (summary data)	Fusion rate 96% at mean follow-up of 15 months	Higher risk of early infections only at >2.1 mg/level; lower doses—no increased risk	[92]
TLIF (one-/ two-level)	0.5–1.0	Prospective multicenter single-arm, n = 30	≈98% fusion	No serious complications reported, including heterotopic ossification (HO)	[30]
PLF (lumbar)	Not specified (controlled doses)	RCT, *n* = 74	Higher fusion quality and lower nonunion rate vs. AIBG	No serious complications reported	[72]
PLF (lumbar)	Various (pooled data)	Meta-analysis of 14 RCTs (789 rhBMP-2 vs. 727 ICBG)	Higher fusion rates; reduced operative time and blood loss	Fewer reoperations related to donor site morbidity	[32]
Deformity surgery (ASD)	Tendency to decrease: ~26.6 → 20.7	Observational data over a decade (2008–2018)	Stable nonunion rates at lower doses	Nearly two-fold reduction in complications requiring reoperation	[57]

### 7.2. Modern Carriers for Controlled and Safe Delivery of rhBMP-2

Although traditional collagen carriers are simple and widely used, as noted earlier, they are often associated with rapid burst release and a high incidence of side effects. This has led to the development of modern composite scaffolds that combine bioceramic frameworks with polymers such as poly(lactic-co-glycolic acid) (PLGA) and bioactive components like nanohydroxyapatite (nHA) with chitosan microspheres and poly-l-lactic acid (PLLA), which provide a more controlled and spatially stable rhBMP-2 release, thereby enhancing osteoinductivity while reducing complication risks. Of particular interest are CaS/HA hydrogels and composite dual-release hydrogels, which can combine regenerative therapy with antimicrobial protection, increasing their clinical potential in dental and orthopedic surgery. For instance, Song et al. demonstrated a reliable sequential release system for dental implantation, in which vancomycin was released over the first two days, followed by sustained rhBMP-2 release for approximately 12 days. This strategy improved osseointegration while simultaneously preventing infection and inflammation [93]. The promise of bioceramic scaffolds has been further confirmed in experimental studies. Cho et al. reported that a 3D-printed CaO–SiO_2_–P_2_O_5_–B_2_O_3_ scaffold loaded with rhBMP-2 achieved complete healing of critical femoral bone defects in rabbits: 100% bone union in the rhBMP-2 group versus only 29% in controls, with nearly doubled bone volume and significantly enhanced mechanical strength [94]. Alongside such single-phase designs, multiphasic release systems have also emerged, combining synthetic polymers such as poly(lactic acid) (PLA), PLGA, and polyethylene glycol (PEG) copolymers with bioceramics (β-TCP, HA, and BCP). A notable example is the work of Lee et al., who developed a collagen–HA scaffold encapsulating PLGA and alginate microspheres. This system enabled phased release of rhBMP-2 and bisphosphonate, resulting in enhanced osteogenesis while simultaneously reducing bone resorption [92].

Hydrogel-based composites likewise demonstrate strong potential. Deng et al. tested a bio–hydroxyapatite–β-TCP hydrogel loaded with rhBMP-2, which successfully promoted bone regeneration in vivo in a nonunion model [95]. In another study, rhBMP-2 was incorporated into chitosan nanoparticles embedded into a 3D PLGA/nHA framework, achieving less than 10% release within the first 48 h and up to 61% over 30 days. In a rabbit mandibular defect model, this system produced 45% bone regeneration after 12 weeks, compared with only 19% in controls [96]. Similarly, Hong et al. introduced a 3D-printed PLLA scaffold conjugated with rhBMP-2, which displayed strong osteoinductive potential while maintaining surgical ease of use in a mouse calvarial defect model [97]. Supporting clinical data also suggest that HA granules can act as stable carriers: in a retrospective spinal fusion study, rhBMP-2 delivered via HA granules improved bone fusion quality and density without increasing complication rates, compared with standard techniques [72].

Beyond synthetics, biopolymeric matrices have attracted significant interest due to their biocompatibility, biodegradability, and ability to modulate the inflammatory response. Materials such as gelatin, HA, chitosan, alginate, fibrin, and fibronectin are being actively explored as platforms for localized rhBMP-2 delivery. Gelatin, as a collagen derivative, can influence the mode of regeneration, while its combination with PLGA improves scaffold stability. Hybrid systems, such as collagen/β-TCP or PLGA/alginate composites, are particularly promising: Xia et al. demonstrated that these carriers reduced the required therapeutic rhBMP-2 dose by 5–10-fold, extended release duration to 21 days, and improved bone morphology, increasing mineral density by 34% compared with controls [98]. Similarly, Li et al. developed a bioactive glass modified with copper and strontium ions, loaded with rhBMP-2. In a rat model of infected long bone defects, this injectable hydrogel provided prolonged rhBMP-2 release while simultaneously suppressing inflammation, stimulating angiogenesis, and accelerating bone regeneration. Importantly, the matrix reduced pro-inflammatory cytokine expression (TNF-α, IL-1β) while upregulating angiogenic and osteogenic markers, and biodegraded completely without residual fibrosis or chronic inflammation [99]. Comparable outcomes were observed with alginate–chitosan composites. Rhee et al. showed that these scaffolds regulated the phases of the inflammatory response, inducing moderate mononuclear infiltration on day one, followed by rapid switching to an M2 macrophage phenotype, with reduced pro-inflammatory mediators and enhanced tissue remodeling. Such matrices not only enable localized rhBMP-2 delivery but also help create an immunoprivileged microenvironment favorable for bone regeneration [100].

Efforts to improve carrier performance have also focused on systems capable of long-term, stable release without harmful concentration spikes. One such approach involves PLGA/TiO_2_ nanotubes integrated with rhBMP-2. In a pilot study, Zhang et al. showed that PLGA-coated TiO_2_ nanotubes provided controlled rhBMP-2 release and enhanced osseointegration of dental implants in vivo [101]. Extending this concept, Xu et al. developed mineralized microparticles encapsulated in a chitosan/PEG hydrogel, capable of releasing rhBMP-2 for up to 75 days. The system demonstrated high biocompatibility, enhanced angiogenesis, and robust bone regeneration in a rat large defect model, with minimal burst release (<30% of rhBMP-2 over 75 days) [102]. Likewise, nanocapsule-based delivery systems responsive to matrix metalloproteinases have shown promise in providing site-specific, controlled protein release while maintaining structural and functional activity. Studies demonstrated that both simultaneous and sequential delivery of BMP2 and SDF-1α promoted bone repair more effectively than BMP2 alone, without inducing ectopic bone formation due to low circulating factor concentrations and sustained nanocapsule-mediated release [103]. This strategy may be particularly valuable in osteoporotic patients, for whom invasive interventions are limited and intrinsic regenerative potential is compromised. For example, Castillo-Santaella et al. described PLGA nanoparticles (100–500 nm) stabilized with Pluronic F68 and encapsulating rhBMP-2. This system exhibited a low burst release profile and robust stimulation of osteogenic mesenchymal stem cell (MSC) differentiation in vitro, as well as improving protein stability with a granular release pattern [104].

In summary, accumulated preclinical and clinical evidence indicates that the choice of delivery system is a decisive factor in both the efficacy and safety of rhBMP-2-based bone regeneration. Traditional collagen sponges, while historically instrumental for clinical translation, are characterized by rapid burst release kinetics, which frequently lead to radiographically detectable heterotopic ossification and early osteolytic changes, particularly when high or poorly contained doses are applied. Although these complications are often subclinical, their occurrence underscores the biological liability of unmodulated release. In contrast, ceramic and composite carriers that provide slower and more sustained release allow for lower total rhBMP-2 doses without loss of regenerative effect, achieve higher or equivalent fusion rates, and markedly reduce the risk of inflammatory reactions, ectopic ossification, and early bone resorption. A comparative summary of rhBMP-2 efficacy across various delivery carriers is provided in Table 3.

## 8. Molecularly Oriented Strategies for the Delivery of Osteogenic Factors

### 8.1. Chemically Modified RNA

One of the most promising developments in this field is the use of cmRNA, which consists of synthetic transcripts with nucleotide substitutions designed to improve molecular stability while reducing innate immune activation [105]. Recent studies have demonstrated that cmRNA encoding BMP2 can serve as an efficient and safe platform for growth factor delivery, in several respects surpassing recombinant protein therapy. This strategy allows for localized BMP2 gene expression strictly at the site of interest, thereby reducing systemic risks. For example, De La Vega et al. applied an optimized BMP2 cmRNA delivered via a nanomaterial-based carrier into a rat critical-sized femoral defect. The treatment resulted in complete defect closure within only four weeks, without the excessive callus formation typically observed with rhBMP-2. The regenerated bone exhibited near-physiological mechanical properties, and remodeling began earlier than in the protein-treated groups. Interestingly, while the process advanced through hypertrophic chondrocytes indicated by type X collagen expression, it did not proceed to full ossification, underscoring the dose-sensitive nature of cmRNA-mediated regeneration. In contrast, rhBMP-2 treatment did not induce type II or type X collagen, indicating a failure to initiate the chondrogenic cascade. The authors further observed that cmRNA-induced osteogenesis predominantly proceeded via endochondral ossification, in contrast to the less physiological intramembranous pathway activated by rhBMP-2 [106].

These findings are consistent with other experimental models. Jriyasetapong et al. encapsulated N1-methylpseudouridine-modified messenger RNA (mRNA) into lipid nanoparticles and implanted this system into rat femoral bone. Remarkably, even a minimal dose of 5 μg induced substantial increases in bone volume, trabecular thickness, and titanium implant integration, with outcomes comparable with those observed at 15 μg. This suggests that effective regeneration can be achieved with lower therapeutic doses, reducing the treatment burden without compromising the efficacy [107]. Efforts to achieve more prolonged and localized release have also turned toward exosome-based systems. Yang et al. employed exosomes engineered with a CP05 peptide capable of specifically binding to CD63. These vesicles were covalently integrated into a gelatin methacrylate (GelMA) hydrogel, creating a GelMA-CP05 construct that provided sustained release of BMP2 cmRNA-loaded exosomes directly at the defect site. In a rat calvarial defect model, this system promoted uniform bone formation without the hyperostosis often associated with rhBMP-2. The hydrogel maintained structural stability during the repair and fully degraded after reconstruction, without inducing inflammatory responses [108].

Alternative strategies that combine in vivo and ex vivo approaches have also shown promise. Surisaeng et al. compared the direct delivery of BMP2 mRNA via lipid nanoparticles with the transplantation of bone marrow mesenchymal stem cells (BMSCs) pre-transfected with BMP2 mRNA. The ex vivo approach produced the most pronounced regenerative outcomes, with BMSCs functioning not as structural elements but rather as local, transient sources of BMP2 expression. Notably, the cells themselves did not undergo osteogenic differentiation, highlighting the importance of paracrine signaling and the feasibility of using cells as vector systems for temporary protein delivery [109]. These studies demonstrate that cmRNA encoding BMP2 represents a highly promising platform for osteogenic therapy, combining transient expression with precise dose control and a lack of genomic integration risk. By enabling osteogenesis through the more physiological endochondral pathway and reducing complications typical of protein-based treatments, cmRNA technologies mark a logical step forward in the evolution of growth factor delivery strategies, bridging protein- and vector-based approaches. The contrasting approaches of regional gene therapy and conventional rhBMP-2 delivery for bone regeneration are illustrated in Figure 4.

### 8.2. Regional Gene Therapy as a Strategy for Sustained In Vivo BMP2 Expression

Another rapidly advancing direction is regional gene therapy, aimed at achieving stable local expression of BMP2 in vivo. This approach provides a more prolonged and physiologically relevant stimulus, promoting chondrogenesis, suppressing fibrotic processes, and creating an anti-inflammatory tissue microenvironment essential for successful bone regeneration. One of the earliest demonstrations of this strategy was reported in a study employing human umbilical cord blood-derived mesenchymal stem cells (UCB-MSCs) transduced with a lentivirus encoding BMP-2 (LV-BMP-2) and implanted into murine subcutaneous muscle pockets. By the fourth week, robust heterotopic bone formation was observed at the implantation site, confirming the effectiveness of ex vivo gene delivery [110]. Subsequent studies provided deeper mechanistic insights. Sarkar et al. compared the expression profiles of hematoma cells forming within rat femoral critical-sized defects treated with either rhBMP-2 or LV-BMP-2-transduced MSCs. Using scRNA-seq, the authors showed that gene therapy resulted in an increased proportion of chondrocytes and a reduction in fibroblasts and pro-inflammatory macrophages, suggesting a more balanced and modulated osteogenic response compared with protein delivery [111]. Similarly, Ihn et al. demonstrated the therapeutic potential of ex vivo gene therapy in correcting femoral defects in animals through implantation of human BMSCs transduced with LV-BMP-2 and seeded on compression-resistant matrices. Complete defect healing was achieved with both standard and high cell doses (5 × 10^6^ and 1.5 × 10^7^), although the higher dose was associated with an elevated risk of heterotopic ossification. These findings illustrate the strong regenerative potential of lentiviral BMP-2 delivery while also highlighting dose-related safety concerns [112].

Further technological refinement has been achieved by integrating gene-modified cells with advanced biomaterial fabrication. Lin et al. combined LV-BMP-2-transduced human BMSCs with 3D bioprinting of biocompatible hydrogels using visible light projection stereolithography (VL-PSL). This allowed for the fabrication of anatomically precise constructs with homogeneously distributed cells. Functionally, the system provided stable BMP2 expression for up to 56 days in vitro, with sustained osteogenic differentiation. In vivo experiments using a severe combined immunodeficiency (SCID) mouse subcutaneous implantation model showed dense bone formation within 14 days, progressing to mature trabecular bone with robust vascularization by days 28–56. GFP labeling confirmed the presence of transplanted LV-BMP-2 BMSCs at both the periphery and the center of the newly formed bone [113].

Despite these advances, gene therapy platforms raise important biosafety considerations. Although Bell et al. reported that lentiviral-transduced MSCs remained localized at the defect site, with the systemic viral load decreasing to negligible levels by day 84, the use of integrating viral vectors such as retroviruses and lentiviruses carries a risk of insertional mutagenesis, particularly when integration occurs near proto-oncogenes. To address this, safety mechanisms such as suicide gene cassettes have been proposed [114]. Even adeno-associated viruses (AAVs), long considered relatively safe, have recently raised concerns due to potential integration into sensitive genomic loci [115]. Consequently, non-integrating systems are increasingly favored for clinical translation. Among these, plasmid DNA stands out as a promising alternative. Its transient expression avoids chronic tissue overstimulation, does not elicit adaptive immune responses typical of viral proteins, and carries no inherent risk of genomic integration when properly designed [116]. Moreover, for bone regeneration, prolonged BMP2 expression is not necessary; instead, a transient signal is sufficient to trigger osteogenic cascades [117]. Nevertheless, even with plasmid or viral vectors encoding BMP2, ectopic bone formation outside the defect area has been observed, underscoring the need for systems that can provide precise spatiotemporal control of gene expression [118].

Several recent studies exemplify the effort to meet this challenge. Jalal et al. demonstrated that complexing BMP2 plasmids with the engineered GET transfection peptide FGF2BLK158R enabled stable transfection of human MSCs, supporting osteogenic differentiation without genomic integration or immune activation, as confirmed by alkaline phosphatase activity and mineralization assays in vitro. Sun et al. validated a different approach, employing a visible-light crosslinkable GelMA hydrogel loaded with hBMSCs and a recombinant AAV vector encoding human BMP2. In a mouse with SCID calvarial defect model, this combined gene + cell platform promoted localized BMP2 expression, osteogenesis, and spatially precise fixation, while avoiding the need for supraphysiological doses of exogenous protein [119]. Khvorostina et al. developed a gene-activated scaffold (GAS) using 3D cryoprinting of sodium alginate carrying BMP2 plasmid DNA polyplexes with polyethylenimine (PEI). This system enabled in situ transfection of host cells at the defect site, inducing robust osteogenesis in critical-sized femoral defects in Wistar rats [120]. Parallel efforts have explored multi-factor approaches. Walsh et al. created a collagen–HA scaffold functionalized with star-shaped poly(l-lysine) polypeptides carrying plasmids encoding both BMP2 and vascular endothelial growth factor. In a rat calvarial defect model, this dual-delivery system induced six-fold higher bone volume compared with the scaffold alone, by simultaneously promoting osteoinduction and angiogenesis [121]. Similarly, Meglei et al. designed a collagen I–platelet-rich plasma hydrogel incorporating BMP2 plasmid polyplexes, which achieved 37% bone regeneration, compared with only 20% with plasmid-free hydrogel and 2–3% in empty defects. Histological analysis revealed mature trabecular bone with strong vascularization and no signs of inflammation, fibrosis, or heterotopic ossification [122].

Moving toward clinically adaptable solutions, Moncal et al. demonstrated intraoperative bioprinting, enabling direct fabrication of 3D constructs during surgery. Their bioink combined BMP2 plasmid-loaded chitosan nanoparticles with free platelet-derived growth factor subunit B (PDGF-B) plasmids, achieving biphasic release: an initial burst of PDGF-B (~10 days), followed by sustained BMP2 expression (~5 weeks). Implanted into critical calvarial defects in Fischer rats, these constructs achieved nearly 40% new bone volume and >90% defect coverage within six weeks, compared with only ~10% bone formation and 25% coverage in untreated controls [123]. Similarly, Vasiliev et al. employed antisolvent 3D printing to fabricate porous PLGA scaffolds carrying adenoviral BMP2 vectors. Implantation into rat cranial defects induced mature mineralized bone formation along and within the scaffold, as confirmed by micro-computed tomography and histological analysis [124].

Collectively, these platforms share two fundamental advantages: highly localized BMP2 gene delivery, which minimizes the risks of diffusion or ectopic expression, and the integration of mechanical and biological activity into a single construct. These attributes highlight the potential of regional gene therapy to serve as a clinically viable alternative to auto- or allograft transplantation, with opportunities for scaling and customization to patient-specific anatomical requirements.

## 9. Conclusions

In light of the limited regenerative capacity of large bone defects, as well as the clinical and ethical constraints of autologous transplantation, contemporary biomedical research is increasingly focused on developing effective and safe alternative strategies for skeletal repair. Among the most extensively investigated osteoinductive agents, rhBMP-2 remains one of the most studied, as it possesses a strong capacity to stimulate bone formation. And yet, its widespread clinical use continues to be challenged by the need to optimize recombinant production methods, enhance protein stability, and precisely regulate its bioavailability.

These challenges necessitate not only careful molecular refinement of the protein itself but also a rational design of delivery matrices capable of providing a controlled and localized release. The expanding range of carriers available today, from traditional collagen scaffolds to sophisticated bioceramic–polymeric composites, has already made it possible to reduce the frequency of adverse effects and improve therapeutic outcomes by offering spatiotemporal control over rhBMP-2 release.

In parallel, a new generation of molecularly oriented strategies has emerged, including cmRNA and regional gene therapy, both of which enable transient in vivo BMP2 expression while fostering a pro-regenerative microenvironment.

At the same time, it is increasingly clear that the clinical value of rhBMP-2 depends on careful, case-specific application. Across trials, rhBMP-2 typically improves fusion rates by only single-digit absolute percentages, meaning its use should be reserved for situations where that modest benefit outweighs the risks and the costs. Emerging delivery technologies, such as hydrogels, nanocomposites, and protein-based scaffolds, are beginning to tip this balance by achieving similar bone regeneration with lower doses and fewer complications. Still, high manufacturing costs, complex regulatory approval pathways, and scalability remain major barriers to broader clinical adoption. To move forward, future rhBMP-2-based therapies must balance potency and safety, demonstrate cost-effectiveness, and align with clearly defined clinical needs. By emphasizing evidence-based risk–benefit tradeoffs, patient-specific strategies, and translational feasibility, next-generation delivery systems can help make rhBMP-2 therapies both practical and clinically impactful.

## Figures and Tables

**Figure 1 ijms-26-10723-f001:**
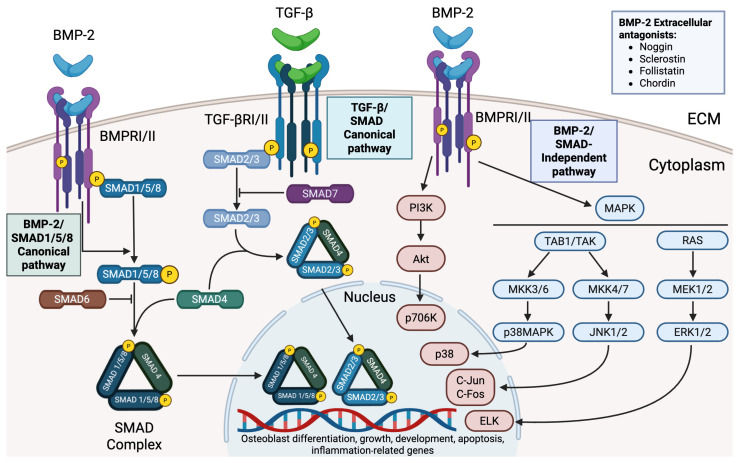
Canonical and non-canonical BMP-2 signaling in osteogenic regulation. BMP-2 binds a heterotetrameric complex of type I and type II BMP receptors (BMPR-I/II), triggering SMAD1/5/8 phosphorylation and SMAD4 complex formation with subsequent nuclear translocation; this canonical SMAD pathway is tightly regulated by extracellular BMP antagonists that sequester BMP-2. In parallel, BMP-2 activates SMAD-independent cascades, including MAPK pathways (via TAK1, leading to p38/JNK/ERK activation) and PI3K/Akt signaling, which stimulate downstream transcription factors such as AP-1 (c-Fos/c-Jun) and Elk-1 to induce osteogenic and inflammatory genes. These combined pathways promote osteoblast differentiation and bone formation, while also influencing cell proliferation, inflammatory responses, and apoptosis. TGF-β signaling (via SMAD2/3 activation) is depicted in the diagram for comparison of pathway specificity. Created in Biorender. Ruchko E. (2025) https://app.biorender.com/illustrations/689b385c4a21dd0068f29b30.

**Figure 2 ijms-26-10723-f002:**
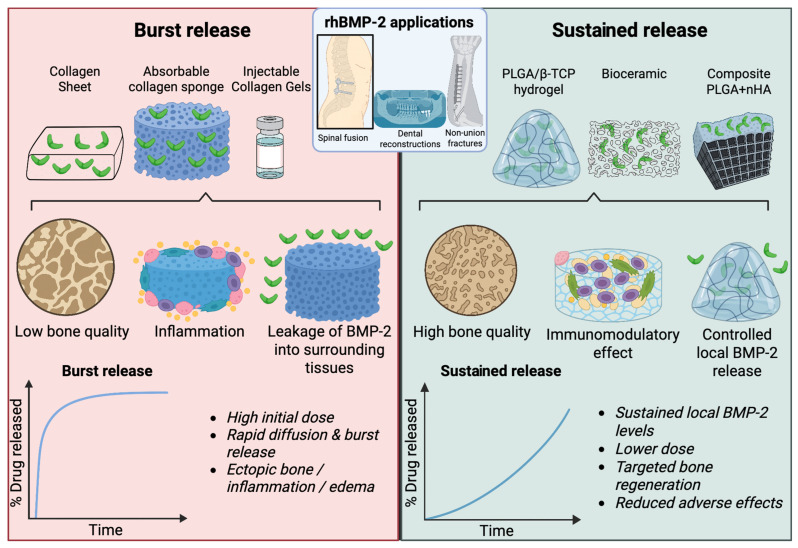
Comparison of burst release versus sustained release delivery of rhBMP-2. Traditional collagen-based carriers (e.g., collagen sheets, absorbable sponges, and injectable gels) typically cause a high initial burst release of rhBMP-2, leading to rapid diffusion from the defect, increased risk of inflammation, ectopic ossification, and poor-quality bone formation. In contrast, modern controlled release systems (e.g., PLGA/β-TCP hydrogels, bioceramics, and composite PLGA + HA scaffolds) provide sustained local BMP-2 release, support an immunomodulatory microenvironment, enhance bone quality, and reduce adverse effects. The lower panels illustrate the characteristic release kinetics for each approach. The schematic was created in Biorender. Ruchko E. (2025) https://app.biorender.com/illustrations/68dbb042bde8af980da5518d.

**Figure 3 ijms-26-10723-f003:**
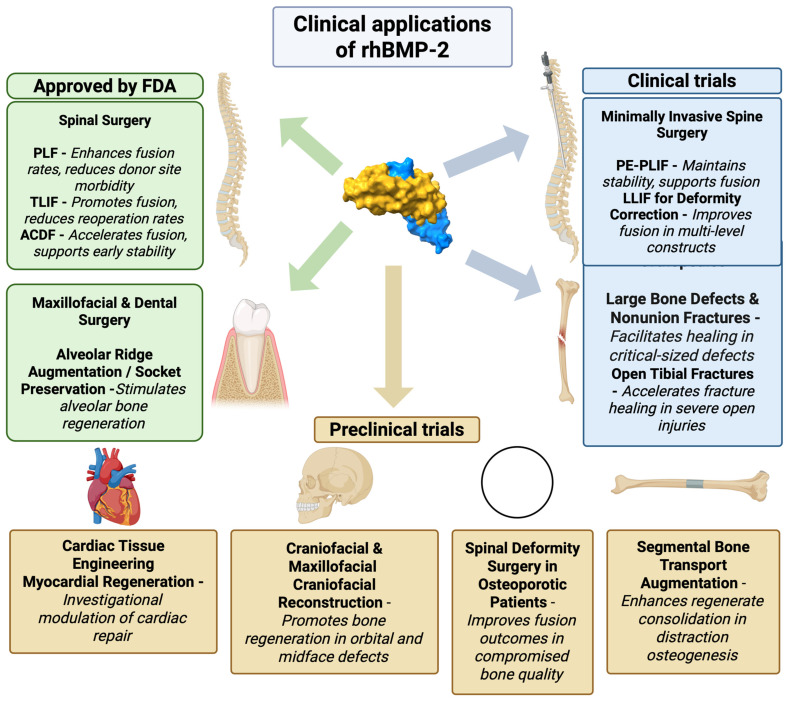
Clinical applications of rhBMP-2. Molecular surface representation of the rhBMP-2 dimer, with each polypeptide chain shown in a different color (chain A in yellow and chain B in blue). Approved indications include spinal fusion surgeries, PLF [32,72], TLIF [61,69,70,73], and ACDF [74,75,76], as well as maxillofacial and dental surgery for alveolar ridge augmentation and socket preservation [30]. Ongoing clinical trials are exploring rhBMP-2 use in minimally invasive spine surgery techniques, such as PE-PLIF [60] and LLIF [71] for deformity correction, as well as in orthopedics for the treatment of large bone defects and nonunion fractures. Preclinical studies are investigating novel applications in cardiac tissue engineering [44,45,46], Craniofacial & Maxillofacial Craniofacial Reconstruction [77], Spinal Deformity Surgery in Osteoporotic Patients [57], and Segmental Bone Transport Augmentation [59]. The schematic was created in Biorender. Ruchko E. (2025) https://app.biorender.com/illustrations/67392de7c5e5325c07fe309a.

**Figure 4 ijms-26-10723-f004:**
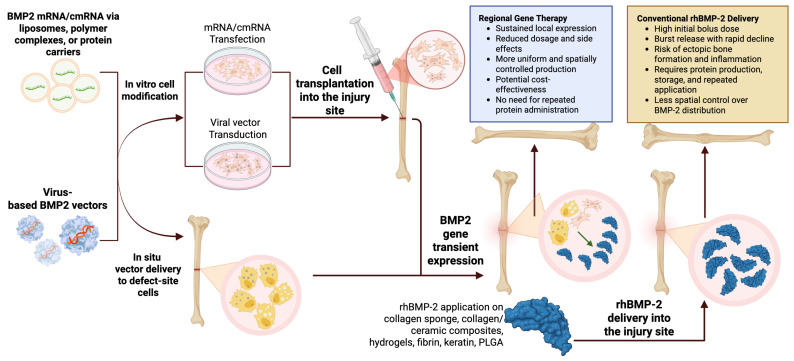
Regional gene therapy versus conventional rhBMP-2 delivery for bone regeneration. Regional gene therapy employs in vitro modification of MSCs or resident cells with BMP2 mRNA/cmRNA or viral vectors, followed by transplantation into the defect, or in situ vector delivery to local cells, providing transient, localized BMP2 expression with lower systemic exposure. In contrast, conventional delivery applies exogenous rhBMP-2 on carriers such as collagen sponges, collagen/ceramic composites, hydrogels, fibrin, keratin, or PLGA, typically requiring high initial doses with burst release, leading to greater risks of ectopic ossification, inflammation, and uneven BMP-2 distribution. The schematic was created in Biorender. Ruchko E. (2025) https://app.biorender.com/illustrations/68dbbdafd579e01e97c57a66.

**Table 3 ijms-26-10723-t003:** Comparison of rhBMP-2 efficacy across different delivery carriers.

Carrier Type	Duration of rhBMP-2 Release	Efficacy	Risk of Complications	Experimental Model	Description
Collagen Sponge	Short (1–3 days, burst release)	High osteoinduction, but unstable bone structure	High (inflammation, ectopic ossification, osteolysis)	In vivo, Clinical	Classical carrier; requires high doses (mg), which increases the incidence of side effects
Hydrogel	Intermediate (7–14 days)	Moderately high osteoinduction, good integration	Medium	In vivo	Can be modified for dual release (e.g., rhBMP-2 + antibiotic)
Bioceramic Scaffold (HA, β-TCP)	Intermediate–long (14–30 days)	High mineralization and mechanical strength	Medium–low	In vivo, Clinical	Provides good osteoconduction and stability; HA granules enhance fusion quality
Composite Scaffold (polymer + ceramic)	Long (up to 21–30 days)	High mineralization, uniform bone formation	Low	In vivo	Allows reduction of rhBMP-2 dose by 5–10 times compared with collagen
3D-Printed Scaffold	Long, controlled (14–28 days)	High osteoinduction and architectural adaptation	Low	In vivo	Enables patient-specific design tailored to the defect
Nanoparticles/ Nanofibers	Very long (up to 60–75 days)	High osteoinduction at low doses	Low	In vivo	Minimal burst release, targeted delivery, and excellent biocompatibility

## Data Availability

No new data were created or analyzed in this study. Data sharing is not applicable to this article.

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
