# Peer review of "Optimizing rhBMP-2 Therapy for Bone Regeneration: From Safety Concerns to Biomaterial-Guided Delivery Systems"

_ijms, 2025, doi:10.3390/ijms262110723_

Round 1

Reviewer 1 Report

Comments and Suggestions for Authors

Recommendation: major revision before acceptance. The paper is promising and timely, but it needs clearer scope and claims, better structure/numbering, tighter handling of a few bold statements (with precise context and citations), and small but numerous editorial fixes.

Major comments and questions:

  1. About scope/claim precision:  In the introduction you write that rhBMP-2 “remains the only FDA-approved bone graft substitute since 2002.” Please clarify this to “the only FDA-approved BMP-based bone graft substitute” (if that is your intent), and limit the claim precisely to the relevant indication(s)/device (INFUSE™/LT-CAGE; P000058). Right now it reads broader than warranted. Could you rephrase and anchor the indication(s) explicitly to the FDA source you cite?
  2. Abstract: Strong, but a closing sentence with one explicit clinical recommendation (e.g., “keep ≤0.7 mg/level when possible in cervical indications”) would help readers.
  3. Section 3 (Expression systems): Good comparative detail; consider a short table listing yields (pg/mL → µg/mL) for CHO, HEK, cell-free, plant, E. coli with pros/cons to improve readability.
  4. Section 3.2 / Table 1: Ensure all outcomes (fusion rates/timelines) and complications have matching references in the table (some cells are summarized; adding [ref] labels in each row improves traceability).
  5. Section 4: Where you note dose reduction over a decade (26.6→20.7 mg/level), please add procedure context and whether this was overall dose or per level (the text says per level—good—just restate in the sentence).
  6. Section 5.2 (Hyperostosis/Osteolysis): Please define timing (early vs. late complications) and carrier/dose wherever possible to guide prevention.
  7. Tables: Consider adding a “Dose needed vs. burst-release risk” column in Table 2; the present table hints at it but an explicit column would sharpen the message.
  8. Section numbering inconsistency:  The manuscript appears to have duplicate section numbers: there is a “5. Patient-Oriented Strategies…” and later another “5. Risks and Adverse Effects…”, followed by “6. Carriers…” and “7. Molecularly Oriented Strategies…”. Please correct the numbering and ensure cross-references (Figure 1, Tables) still resolve. (See sections around the transition from Figure 1 to “Risks…”.)
  9. Hyphenation/line breaks: Several words are broken at line ends (e.g., “pseudar-throsis”, “integra-tion”). Please run a final proof to remove hyphenation artifacts.
  10. Units/decimals: Standardize µg/mL (not “ug/mL”), mg/level, and decimal points (0.5 not 0,5 unless IJMS requests comma).
Comments on the Quality of English Language

See my report

Author Response

Response to Reviewer X Comments

1. Summary

We sincerely appreciate your careful reading of our manuscript and the encouraging remarks. Thank you for recognizing the timeliness and potential of our work. We are grateful for your thorough and constructive feedback, which has provided us with clear direction for improving the paper.

We have carefully considered all your major comments and questions and have implemented revisions accordingly. The scope and claims have been clarified, the structure and numbering have been corrected, bold statements have been tempered with precise context and citations, and numerous editorial fixes have been made throughout the manuscript.

Point-by-point responses to your specific recommendations are provided below.

All revisions in the manuscript have been clearly marked: major new or substantially revised paragraphs are highlighted in green, while smaller textual edits are underlined in the track-changes mode for ease of review.

2. Questions for General Evaluation

Reviewer’s Evaluation

Response and Revisions

Is the work a significant contribution to the field?

[Please give your response if necessary. Or you can also give your corresponding response in the point-by-point response letter. The same as below]

Is the work well organized and comprehensively described?

Is the work scientifically sound and not misleading?

Are there appropriate and adequate references to related and previous work? 

Is the English used correct and readable?        

3. Point-by-point response to Comments and Suggestions for Authors

Comments 1: Abstract: Strong, but a closing sentence with one explicit clinical recommendation (e.g., “keep ≤0.7 mg/level when possible in cervical indications”) would help readers.

Response 1: We are grateful to the reviewer for the valuable feedback. We have revised the Abstract to include the suggested clinical recommendation.

Lines

It was originally in the text

Corrections

21-23

-

It is noteworthy, that low doses of rhBMP-2 (0.5-0.7 mg/level in anterior cervical discectomy and fusion (ACDF) or 0.5–1.0 mg/level in transforaminal lumbar interbody fusion (TLIF)) provide the optimal balance of efficacy and safety.

Comments 2: Section 3 (Expression systems): Good comparative detail; consider a short table listing yields (pg/mL → µg/mL) for CHO, HEK, cell-free, plant, E. coli with pros/cons to improve readability.

Response 2: We thank the reviewer for this valuable feedback. We did consider creating a summary table with yields and pros/cons for each system. However, given the addition of other new tables and figures throughout the manuscript, we decided to retain the information in the text to avoid overloading the article and to ensure a balanced presentation.

Comments 3: Section 3.2 / Table 1: Ensure all outcomes (fusion rates/timelines) and complications have matching references in the table (some cells are summarized; adding [ref] labels in each row improves traceability).

Response 3: We have verified the table and confirm its accuracy. The table was designed to provide a clear, consolidated overview of the information discussed in the corresponding chapter. Each row in the table relates to a specific type of study, and the reference for each row is provided in the last column of the table. The studies are also discussed in the main body of the chapter.

Comments 4: Section 4: Where you note dose reduction over a decade (26.6→20.7 mg/level), please add procedure context and whether this was overall dose or per level (the text says per level—good—just restate in the sentence).

Response 4: We thank the reviewer for this insight. We have revised the text accordingly.

Lines

It was originally in the text

Corrections

346-349

For instance, Bannwarth et al. reported that the mean dose of rhBMP-2 per surgical level decreased from 26.6 to 20.7 mg (p < 0.001), which was associated with nearly a twofold reduction in complication rates requiring reoperation.

For instance, Bannwarth et al. reported that the mean dose per patient of rhBMP-2 for adult spinal deformity (ASD) surgery decreased from 26.6 to 20.7 mg (p < 0.001), which was associated with nearly a twofold reduction in complication rates requiring reoperation.

Comments 5: Section 5.2 (Hyperostosis/Osteolysis): Please define timing (early vs. late complications) and carrier/dose wherever possible to guide prevention.

Response 5: We thank the reviewer for this important suggestion. We have revised the manuscript to define the timing of hyperostosis and osteolysis and have included more information wherever possible, as recommended.

Lines

It was originally in the text

Corrections

303-304

-

These radiographic findings typically become apparent within the first 3–6 months postoperatively [42,50,51].

313-323

-

This clinically significant hyperostosis is often a late complication, manifesting 6–12 months after surgery, and its risk correlates with higher doses (e.g., >2.0 mg per level in cervical spine) and the use of certain collagen-based carriers that allow for uncontrolled diffusion [50,53].

330-332

-

Osteolysis is predominantly an early complication, peaking at 2–6 weeks postoperatively and often stabilizing by 3–6 months [53,55].

333-334

-

The severity of osteolysis is strongly dose-dependent, with higher doses being major risk factors[55,56].

Comments 6: Tables: Consider adding a “Dose needed vs. burst-release risk” column in Table 2; the present table hints at it but an explicit column would sharpen the message.

Response 6: We thank the reviewer for this suggestion. However, due to the high variability in the literature regarding dosing and release kinetics, we have kept the current table format to avoid an oversimplified or misleading comparison.

Comments 7: Section numbering inconsistency:  The manuscript appears to have duplicate section numbers: there is a “5. Patient-Oriented Strategies…” and later another “5. Risks and Adverse Effects…”, followed by “6. Carriers…” and “7. Molecularly Oriented Strategies…”. Please correct the numbering and ensure cross-references (Figure 1, Tables) still resolve. (See sections around the transition from Figure 1 to “Risks…”.)

Response 7: We have carefully corrected all section and subsection numbers throughout the manuscript and verified that all cross-references to tables and figures resolve correctly. Moreover we have reorganized the sections to improve logical flow and the structure of the manuscript.

Line

It was originally in the text

Corrections

339

4. New Clinical Applications of rhBMP-2

4. Risks and Adverse Effects of rhBMP-2 Therapy

478

5. Patient-Oriented Strategies for rhBMP-2 Application

5. New Clinical Applications of rhBMP-2

201

5. Risks and Adverse Effects of BMP-2 Therapy

6. Patient-Oriented Strategies for rhBMP-2 Application

567

6. Carriers as Critical Determinants of rhBMP-2 Efficacy and Safety

7. Carriers as Critical Determinants of rhBMP-2 Efficacy and Safety

758

7. Molecularly Oriented Strategies for the Delivery of Osteogenic Factors

8. Molecularly Oriented Strategies for the Delivery of Osteogenic Factors

924

5. Conclusions

9. Conclusions

Comments 8: Hyphenation/line breaks: Several words are broken at line ends (e.g., “pseudar-throsis”, “integra-tion”). Please run a final proof to remove hyphenation artifacts.

Response 8: We have carefully proofread the manuscript to eliminate all incorrect hyphenations at line breaks.

Comments 9: Units/decimals: Standardize µg/mL (not “ug/mL”), mg/level, and decimal points (0.5 not 0,5 unless IJMS requests comma).

Response 9: We have carefully reviewed the manuscript and standardized all units and decimals according to the required format.

Line

It was originally in the text

Corrections

565 -
Table 1. Optimization of rhBMP-2 dosing in clinical studies.

Row 1

Row 5

0,5

0,5-1,0

0.5

0.5-1.0

4. Response to Comments on the Quality of English Language

Point 1:

Response 1:    We thank the reviewer for the suggestion to improve the clarity of the English language. We have thoroughly reviewed the manuscript and made edits to enhance readability and phrasing where necessary.

5. Additional clarifications

We are sincerely grateful to the reviewer for the constructive and thoughtful feedback. The revisions have, in our view, substantially strengthened the manuscript, enhancing its logical structure, mechanistic depth, and practical relevance. It is our hope that the changes satisfactorily address all concerns and that the revised manuscript will serve as a valuable reference for both researchers and clinicians.

We look forward to the editorial decision on the revised submission and remain available to address any further questions or comments that may arise. Our sincere thanks are also extended to the editors of the International Journal of Molecular Sciences for their careful consideration and support throughout the review process.

With kind regards,

The team of authors.

Reviewer 2 Report

Comments and Suggestions for Authors

This review addresses a clinically significant topic and provides a timely overview of rhBMP-2 delivery strategies. However, the analysis lacks depth in critically evaluating recent advances and their mechanistic foundations. The logical flow could also be strengthened. With revisions emphasizing mechanistic insight and comparative discussion, this review could become a more valuable reference for future research and application.
Comments for the Authors:
The following points require careful attention and substantial revision to enhance the manuscript's logical flow, depth, accuracy, and overall impact:

1.    The current organization of sections introduces some logical interruptions that could affect readability and argumentative coherence.
1)    It is advised to reposition "5. Risks and Adverse Effects of BMP-2 Therapy" to follow directly after "3. Expression Systems... ". Placing the discussion of safety implications prior to the presentation of optimization strategies (Section 6 and 7) would better contextualize the necessity of such advancements.
2)    Additionally, "3.2. Optimization of rhBMP-2 Dose and Delivery" could be seamlessly integrated into "6. Carriers as Critical Determinants... ". Since dosage and carrier design are fundamentally interrelated, combining these sections would avoid redundancy and strengthen the narrative continuity.
2.    The introduction or biological foundations section should be expanded to provide a clear rationale for the specific focus on BMP-2 within the broader BMP family. A explicit distinction must also be made between native BMP-2 and recombinant human BMP-2 (rhBMP-2), highlighting key differences in structure, function, and clinical applicability. Including a concise discussion on the structure-function relationship of rhBMP-2 would offer important conceptual clarity.
3.    A critical formatting issue must be resolved: there are currently three different sections all numbered as Section 5 ("Patient-Oriented Strategies...", "Risks and Adverse Effects...", and "Conclusions"). All chapter headings and numbering must be meticulously checked and corrected throughout the manuscript for consistency.
4.    The section "5. Patient-Oriented Strategies for rhBMP-2 Application" provides an excellent opportunity to enhance translational relevance. We recommend incorporating a discussion on tailoring rhBMP-2 dosage and delivery systems according to specific clinical contexts and patient-specific variables. A summary table comparing these factors could significantly increase the section’s utility for clinical readers.
5.    The manuscript contains inconsistent usage of "rhBMP-2" and "BMP2". It is crucial to maintain precise and consistent terminology throughout. "rhBMP-2" is the standard term for the recombinant protein and should be used consistently in clinical and therapeutic contexts (e.g., in the title of section “5. Risks and Adverse Effects of BMP-2 Therapy” should be “...rhBMP-2 Therapy”). "BMP-2" typically refers to the native protein.
6.    The inclusion of high-quality, original schematic figures is highly encouraged to synthesize complex information and significantly improve readability.
1)    For Section 2, consider adding a mechanism-based figure illustrating BMP-2 signaling and its role in osteogenesis.
2)    For Section “5. Risks and Adverse Effects of BMP-2 Therapy”, a comparative diagram contrasting traditional burst release versus contemporary controlled-release profiles would help highlight safety improvements.
3)    For Section 7, a schematic diagram explaining the novel mechanisms of action for cmRNA and regional gene therapy strategies is essential for helping readers grasp these advanced concepts.

Author Response

Response to Reviewer X Comments

1. Summary

Dear Reviewer,

We sincerely appreciate your thorough evaluation of our manuscript and your encouraging remark that the topic is clinically significant and timely. We are especially grateful for your constructive recommendations on improving the depth of mechanistic discussion, logical flow, and overall impact of the review. Your insights have been invaluable in strengthening both the scientific rigor and the readability of the paper.

Point-by-point responses to your specific recommendations are provided below.

All revisions in the manuscript have been clearly marked: major new or substantially revised paragraphs are highlighted in green, while smaller textual edits are underlined in the track-changes mode for ease of review.

2. Questions for General Evaluation

Reviewer’s Evaluation

Response and Revisions

Is the work a significant contribution to the field?

[Please give your response if necessary. Or you can also give your corresponding response in the point-by-point response letter. The same as below]

Is the work well organized and comprehensively described?

Is the work scientifically sound and not misleading?

Are there appropriate and adequate references to related and previous work? 

Is the English used correct and readable?        

3. Point-by-point response to Comments and Suggestions for Authors

Comments 1: 1. The current organization of sections introduces some logical interruptions that could affect

readability and argumentative coherence.

1.1) It is advised to reposition "5. Risks and Adverse Effects of BMP-2 Therapy" to follow directly

after "3. Expression Systems... ". Placing the discussion of safety implications prior to the

presentation of optimization strategies (Section 6 and 7) would better contextualize the necessity

of such advancements.

2) Additionally, "3.2. Optimization of rhBMP-2 Dose and Delivery" could be seamlessly

integrated into "6. Carriers as Critical Determinants... ". Since dosage and carrier design are

fundamentally interrelated, combining these sections would avoid redundancy and strengthen the

narrative continuity.

Response 1: 1.1) Reorganization for logical flow: As advised, we have moved the section “Risks and Adverse Effects of rhBMP-2 Therapy” to immediately follow “Expression Systems…”, so that the safety considerations precede the discussion of optimization strategies. We have also integrated “Optimization of rhBMP-2 Dose and Delivery” into “Carriers as Critical Determinants…”, thereby improving narrative continuity and avoiding redundancy.

Line

It was originally in the text

Corrections

201

Chapter 5. Risks and Adverse Effects of BMP-2 Therapy

Chapter 4. Risks and Adverse Effects of rhBMP-2 Therapy

1.2) We have integrated the subsection ‘Optimization of rhBMP-2 Dose and Delivery’ into the section on ‘Carriers as Critical Determinants of rhBMP-2 Efficacy and Safety’, as dosage and carrier design are indeed interrelated.

Line

It was originally in the text

Corrections

621

Chapter 3.2. Optimization of rhBMP-2 Dose and Delivery 

Chapter 7.1.  Optimization of rhBMP-2 Dose and Delivery

Comments 2: 2. The introduction or biological foundations section should be expanded to provide a clear

rationale for the specific focus on BMP-2 within the broader BMP family. A explicit distinction

must also be made between native BMP-2 and recombinant human BMP-2 (rhBMP-2),

highlighting key differences in structure, function, and clinical applicability. Including a concise

discussion on the structure-function relationship of rhBMP-2 would offer important conceptual

clarity.

Response 2: 2) We have expanded the Introduction to clearly explain why BMP-2 was selected as the focus among the BMP family, and we now include a concise discussion of structural and functional distinctions between native BMP-2 and recombinant human BMP-2 (rhBMP-2), emphasizing their respective roles in clinical application.

In the revised Introduction, we have:

  • Expanded the background on the BMP family, explaining the clinical preference for BMP-2 due to its potent osteoinductive activity.
  • Clearly distinguished between native BMP-2 and rhBMP-2, highlighting differences in structure, glycosylation status, stability, and clinical formulation.
  • Added a brief note on the structure-function relationship of rhBMP-2, emphasizing that the recombinant dimer retains the osteoinductive properties of the native ligand but requires supraphysiologic doses in current delivery formats.

Lines

It was originally in the text

Corrections

61-94

At the cellular level, BMP-2 exerts its effects through binding to type I and type II receptors, activating both Smad-dependent and Smad-independent signaling cascades.

Within the BMP family, which comprises nearly 20 structurally related members, BMP-2 and BMP-7 are recognized as the most potent osteoinductive factors [3]. Among these, BMP-2 exhibits the highest and most reproducible capacity to induce endochondral ossification and bone regeneration. The combination of this pronounced biological activity with its early molecular cloning and successful large-scale recombinant production has made BMP-2 the first BMP to achieve widespread clinical translation. Currently, rhBMP-2 has been evaluated and applied in multiple contexts, including spinal fusion procedures, long bone fracture repair, and maxillofacial augmentation, thereby establishing its role as the benchmark osteoinductive growth factor [3].

Endogenous BMP-2 is synthesized as a precursor protein comprising a signal peptide, a pro-domain, and a mature C-terminal growth factor domain. In the physiological context, the pro-domain facilitates correct folding, dimerization, and secretion, while also regulating latency and extracellular matrix binding. The mature growth factor domain undergoes proteolytic cleavage to release the active dimer, which is sequestered within the bone matrix and becomes bioavailable primarily during bone remodeling or injury [7]. Native BMP-2 therefore acts in a tightly regulated microenvironment, in concert with other matrix-associated molecules such as proteoglycans, growth factors, and matrix metalloproteinases, ensuring spatially restricted and temporally controlled osteogenic signaling [8,9].  

RhBMP-2 is produced in heterologous expression systems, most commonly chinese hamster ovary (CHO) cells or E. coli. Recombinant production yields a purified and standardized protein suitable for therapeutic use. Bacterial-derived variants may offer potential advantages in terms of production efficiency and cost. RhBMP-2, produced in CHO cells, is glycosylated—a modification that can affect its stability and interactions. In contrast, E. coli-derived rhBMP-2 is non-glycosylated but yields high quantities of the active growth factor domain at lower production costs. Despite this difference, both recombinant variants assemble into disulfide-linked homodimers stabilized by a cystine-knot motif [10]. Thus, CHO- and E. coli–derived rhBMP-2 both form biologically active homodimers, but differ in their post-translational modifications and glycosylation, that can influence protein folding, stability, and pharmacokinetics. Clinical investigations indicate that both forms of rhBMP-2 demonstrate osteoinductive efficacy [3,11].

Both native and recombinant BMP-2 signal through heteromeric complexes of type I and type II BMP receptors, activating both Smad-dependent and Smad-independent signaling cascades.

Comments 3: 3. A critical formatting issue must be resolved: there are currently three different sections all

numbered as Section 5 ("Patient-Oriented Strategies...", "Risks and Adverse Effects...", and

"Conclusions"). All chapter headings and numbering must be meticulously checked and corrected

throughout the manuscript for consistency.

Response 3: 3) We thank the reviewer for catching this error. We have carefully corrected all section and subsection numbers throughout the manuscript and verified that all cross-references to tables and figures resolve correctly.

Line

It was originally in the text

Corrections

478

5. Patient-Oriented Strategies for rhBMP-2 Application

6. Patient-Oriented Strategies for rhBMP-2 Application

201

5. Risks and Adverse Effects of BMP-2 Therapy

4. Risks and Adverse Effects of rhBMP-2 Therapy

924

5. Conclusions

9. Conclusions

Comments 4: 4. The section "5. Patient-Oriented Strategies for rhBMP-2 Application" provides an excellent

opportunity to enhance translational relevance. We recommend incorporating a discussion on

tailoring rhBMP-2 dosage and delivery systems according to specific clinical contexts and patient-

specific variables. A summary table comparing these factors could significantly increase the

section’s utility for clinical readers.

Response 4: 4) We thank the reviewer for this valuable suggestion, which indeed enhances the translational relevance of the review. In the revised manuscript, we have expanded the section “Patient-Oriented Strategies for rhBMP-2 Application” to emphasize that the optimal dose and delivery system should be tailored to both the surgical indication and the patient’s biological profile. We now include a concise discussion with representative clinical examples, such as open tibial fractures (6–12 mg via collagen sponge), maxillary sinus augmentation (0.75–1.5 mg/mL), post-extraction socket preservation (requiring slightly higher concentrations), and spinal fusion procedures (typically 1.0–1.5 mg/mL per level), to illustrate context-dependent dosing.

Lines

It was originally in the text

Corrections

528-564

-

The optimal rhBMP-2 dose and delivery system should not be considered fixed parameters but rather tailored to the surgical indication and the biological profile of the patient. In long-bone trauma, pivotal randomized trials have shown that in open tibial fractures, a total dose of 6–12 mg (0.75–1.5 mg/mL) applied via an absorbable collagen sponge accelerates fracture union and reduces the need for secondary interventions compared with standard autograft-based care [78]. In the field of oral and maxillofacial surgery, maxillary sinus floor augmentation has consistently demonstrated reliable bone formation with doses of 0.75–1.5 mg/mL, while post-extraction socket preservation benefits from slightly higher concentrations, which translate into greater ridge volume and improved implant stability [79]. For spinal fusion procedures, rhBMP-2 is typically applied at 1.0–1.5 mg/mL per fusion level ; although its use remains off-label in several jurisdictions, clinical evidence supports improved fusion rates and reduced reoperation risk, particularly in multilevel constructs and adult deformity correction [80].

Beyond the surgical indication itself, patient-specific variables play a critical role in determining the most appropriate rhBMP-2 regimen. Factors such as age, metabolic status, bone quality, smoking, and comorbidities (including diabetes or osteoporosis) directly influence osteoinductive responsiveness. For example, experimental and clinical data suggest that higher doses of rhBMP-2 may partially compensate for impaired bone regeneration in a segmental femoral defect model in diabetes mellitus BB Wistar rats [81], while in osteoporotic conditions, slow-release carriers (e.g. ceramic composites, hydrogel matrices, or hybrid scaffolds) help maintain local bioactivity, reduce systemic exposure, and improve graft stability [82]). Moreover, vascular health has emerged as an important stratification parameter, as impaired perfusion limits rhBMP-2-mediated healing despite adequate dosing [83]. These insights highlight that rhBMP-2 therapy is most effective when embedded into a precision medicine framework, where dosage and delivery vehicles are selected not only according to the anatomical site but also to the patient’s biological and clinical profile. Such individualized strategies maximize efficacy while reducing complications such as ectopic bone formation, infection, or inflammatory reactions, thereby reinforcing the translational relevance of rhBMP-2 in modern regenerative practice. Such considerations provide the conceptual basis for tailoring rhBMP-2 therapy in clinical practice. Table 1 summarizes representative studies across diverse surgical indications, highlighting typical dose ranges, delivery vehicles, patient-specific variables, and key outcomes. This comparative overview underscores that rhBMP-2 efficacy is context-dependent and illustrates how adjusting dosage and scaffold selection according to the clinical scenario and patient characteristics can optimize bone regeneration while minimizing complications.

In addition, we have added a new table “Summary of clinical studies using rhBMP-2 for bone regeneration in various indications” that compares recommended dosing ranges, preferred carrier types, and risk considerations across different clinical indications. This table is intended to increase the section’s utility for clinicians.

Table 1 Summary of clinical studies using rhBMP-2 for bone regeneration in various indications.

Comments 5: 5. The manuscript contains inconsistent usage of "rhBMP-2" and "BMP2". It is crucial to

maintain precise and consistent terminology throughout. "rhBMP-2" is the standard term for the

recombinant protein and should be used consistently in clinical and therapeutic contexts (e.g., in

the title of section “5. Risks and Adverse Effects of BMP-2 Therapy” should be “...rhBMP-2

Therapy”). "BMP-2" typically refers to the native protein.

Response 5: 5) We have systematically reviewed the entire manuscript to ensure consistent use of ‘rhBMP-2’ for the recombinant therapeutic protein in all clinical and therapeutic contexts, including section titles (e.g., now “Risks and Adverse Effects of rhBMP-2 Therapy”).‘BMP-2’ is now reserved exclusively for reference to the native protein or general family context.

Lines

It was originally in the text

Corrections

159, 273, 278, 290 573, 610, 755,

757 – Table 3

BMP-2

rhBMP-2

Comments 6: 6. The inclusion of high-quality, original schematic figures is highly encouraged to synthesize

complex information and significantly improve readability.

1) For Section 2, consider adding a mechanism-based figure illustrating BMP-2 signaling and

its role in osteogenesis.

Response 6: 6.1) We agree that schematic figures significantly enhance clarity. We have therefore added a new figure (Figure 1) in Section 2 depicting canonical and non-canonical BMP-2 signaling pathways, including SMAD-dependent and SMAD-independent cascades relevant to osteogenesis.

Figure 1. Canonical and non-canonical BMP-2 signaling in osteogenic regulation.

BMP-2 binds a heterotetrameric complex of type I and type II BMP receptors (BMPR-I/II), triggering SMAD1/5/8 phosphorylation and SMAD4 complex formation with subsequent nuclear translocation; this canonical SMAD pathway is tightly regulated by extracellular BMP antagonists that sequester BMP-2. In parallel, BMP-2 activates SMAD-independent cascades, including MAPK pathways (via TAK1, leading to p38/JNK/ERK activation) and PI3K/Akt signaling, which stimulate downstream transcription factors such as AP-1 (c-Fos/c-Jun) and Elk-1 to induce osteogenic and inflammatory genes. These combined pathways promote osteoblast differentiation and bone formation, while also influencing cell proliferation, inflammatory responses, and apoptosis. TGF-β signaling (via SMAD2/3 activation) is depicted in the diagram for comparison of pathway specificity

Comments 7: 2) For Section “5. Risks and Adverse Effects of BMP-2 Therapy”, a comparative diagram

contrasting traditional burst release versus contemporary controlled-release profiles would help

highlight safety improvements.

Response 7:
6.2) Included a comparative diagram (Figure 2) in the revised Section on ‘Risks and Adverse Effects of rhBMP-2 Therapy’, contrasting traditional burst-release delivery from collagen sponges with controlled-release from modern scaffolds, to illustrate safety improvements.

Figure 2. Comparison of burst-release versus sustained-release delivery of rhBMP-2.

Traditional collagen-based carriers (e.g., collagen sheets, absorbable sponges, injectable gels) typically cause a high initial burst release of rhBMP-2, leading to rapid diffusion from the defect, increased risk of inflammation, ectopic ossification, and poor-quality bone formation. In contrast, modern controlled-release systems (e.g., PLGA/β-TCP hydrogels, bioceramics, composite PLGA+HA scaffolds) provide sustained local BMP-2 release, support an immunomodulatory microenvironment, enhance bone quality, and reduce adverse effects. The lower panels illustrate the characteristic release kinetics for each approach.

Comments 8: 3) For Section 7, a schematic diagram explaining the novel mechanisms of action for cmRNA

and regional gene therapy strategies is essential for helping readers grasp these advanced

concepts.

Response 8:
6.3) Prepared a new schematic (Figure 4) in Section 7 illustrating novel mechanisms of cmRNA-based and regional gene therapy strategies, emphasizing localized transient BMP-2 expression and reduced adverse effects.

Figure 4. Regional gene therapy versus conventional rhBMP-2 delivery for bone regeneration.

Regional gene therapy employs in vitro modification of MSCs or resident cells with BMP-2 mRNA/cmRNA or viral vectors, followed by transplantation into the defect, or in situ vector delivery to local cells, providing transient, localized BMP-2 expression with lower systemic exposure. In contrast, conventional delivery applies exogenous rhBMP-2 on carriers such as collagen sponges, collagen/ceramic composites, hydrogels, fibrin, keratin, or PLGA, typically requiring high initial doses with burst release, leading to greater risks of ectopic ossification, inflammation, and uneven BMP-2 distribution.

4. Response to Comments on the Quality of English Language

Point 1:

Response 1:  We thank the reviewer for the suggestion to improve the clarity of the English language. We have thoroughly reviewed the manuscript and made edits to enhance readability and phrasing where necessary.

5. Additional clarifications

We are sincerely grateful to the reviewer for the constructive and thoughtful feedback. The revisions have, in our view, substantially strengthened the manuscript, enhancing its logical structure, mechanistic depth, and practical relevance. It is our hope that the changes satisfactorily address all concerns and that the revised manuscript will serve as a valuable reference for both researchers and clinicians.

We look forward to the editorial decision on the revised submission and remain available to address any further questions or comments that may arise. Our sincere thanks are also extended to the editors of the International Journal of Molecular Sciences for their careful consideration and support throughout the review process.

With kind regards,

The team of authors.

Reviewer 3 Report

Comments and Suggestions for Authors

Dear authors,

Thank you so much for submitting your paper to the prestigious journal IJMS.

The paper is interesting and I hope that my comments and remarks will be useful in order to increase the quality of the manuscript.

General aspects:

The manuscript presents a review focused on the optimization of rhBMP-2–based regenerative approaches for complex hard tissue defects. The topic is timely and clinically relevant, given the limitations of conventional grafts and the ongoing challenges in achieving predictable bone regeneration. The abstract is well written, structured, and provides a clear rationale for the review. The emphasis on biomaterial-based carriers, advanced scaffolding technologies, and molecular delivery strategies demonstrates novelty and translational significance.

However, the paper also suggests areas that may require deeper elaboration in the full manuscript. Specifically, the safety profile of rhBMP-2 and the comparative effectiveness of delivery systems need to be critically appraised using quantitative data from both preclinical and clinical studies. Furthermore, potential translational barriers such as regulatory approval, cost, and scalability should be addressed to strengthen the practical impact of the review.

In the same time I can state that the tables are well constructed and contain relevant information.

Minor issues:

  1.  The figure is original or requires copyrights?
  2. My piece of advice is to rephrase the Conclusions section in such a way that will reflect the practical/clinical aspect of the current paper. It will increase the impact of the paper.

In conclusion, I appreciate very much the topic, the quality of the paper and the way in which the paper is structured. Congratulations to the authors!

Please receive my warmest regards and best wishes for future projects!

Author Response

Response to Reviewer X Comments

1. Summary

Dear Reviewer,

We sincerely appreciate the your careful reading of our manuscript and the encouraging remarks about its timeliness, clarity of the abstract, and the quality of the tables. We are grateful for the acknowledgment of the novelty and translational significance of our focus on biomaterial-based rhBMP-2 delivery strategies.

Point-by-point responses to your specific recommendations are provided below.

All revisions in the manuscript have been clearly marked: major new or substantially revised paragraphs are highlighted in green, while smaller textual edits are underlined in the track-changes mode for ease of review.

2. Questions for General Evaluation

Reviewer’s Evaluation

Response and Revisions

Is the work a significant contribution to the field?

[Please give your response if necessary. Or you can also give your corresponding response in the point-by-point response letter. The same as below]

Is the work well organized and comprehensively described?

Is the work scientifically sound and not misleading?

Are there appropriate and adequate references to related and previous work? 

Is the English used correct and readable?        

3. Point-by-point response to Comments and Suggestions for Authors

Comments 1: However, the paper also suggests areas that may require deeper elaboration in the full manuscript. Specifically, the safety profile of rhBMP-2 and the comparative effectiveness of

delivery systems need to be critically appraised using quantitative data from both preclinical and

clinical studies

Response 1: We have carefully addressed the points raised by the reviewer. In particular, we expanded the discussion of the safety profile of rhBMP-2, adding quantitative preclinical and clinical data on on effectiveness of delivery systems and key outcomes in Table 1”Summary of clinical studies using rhBMP-2 for bone regeneration in various indications”:

2) We also enhanced the critical appraisal of delivery platforms, highlighting comparative effectiveness and safety among collagen, ceramic, composite, and advanced controlled-release matrices.We thank the reviewer for highlighting these important aspects and providing a more detailed critical appraisal within the main text:

Lines

It was originally in the text

Corrections

528-564

-

The optimal rhBMP-2 dose and delivery system should not be considered fixed parameters but rather tailored to the surgical indication and the biological profile of the patient. In long-bone trauma, pivotal randomized trials have shown that in open tibial fractures, a total dose of 6–12 mg (0.75–1.5 mg/mL) applied via an absorbable collagen sponge accelerates fracture union and reduces the need for secondary interventions compared with standard autograft-based care [78]. In the field of oral and maxillofacial surgery, maxillary sinus floor augmentation has consistently demonstrated reliable bone formation with doses of 0.75–1.5 mg/mL, while post-extraction socket preservation benefits from slightly higher concentrations, which translate into greater ridge volume and improved implant stability [79]. For spinal fusion procedures, rhBMP-2 is typically applied at 1.0–1.5 mg/mL per fusion level ; although its use remains off-label in several jurisdictions, clinical evidence supports improved fusion rates and reduced reoperation risk, particularly in multilevel constructs and adult deformity correction [80].

Beyond the surgical indication itself, patient-specific variables play a critical role in determining the most appropriate rhBMP-2 regimen. Factors such as age, metabolic status, bone quality, smoking, and comorbidities (including diabetes or osteoporosis) directly influence osteoinductive responsiveness. For example, experimental and clinical data suggest that higher doses of rhBMP-2 may partially compensate for impaired bone regeneration in a segmental femoral defect model in diabetes mellitus BB Wistar rats [81], while in osteoporotic conditions, slow-release carriers (e.g. ceramic composites, hydrogel matrices, or hybrid scaffolds) help maintain local bioactivity, reduce systemic exposure, and improve graft stability [82]). Moreover, vascular health has emerged as an important stratification parameter, as impaired perfusion limits rhBMP-2-mediated healing despite adequate dosing [83]. These insights highlight that rhBMP-2 therapy is most effective when embedded into a precision medicine framework, where dosage and delivery vehicles are selected not only according to the anatomical site but also to the patient’s biological and clinical profile. Such individualized strategies maximize efficacy while reducing complications such as ectopic bone formation, infection, or inflammatory reactions, thereby reinforcing the translational relevance of rhBMP-2 in modern regenerative practice. Such considerations provide the conceptual basis for tailoring rhBMP-2 therapy in clinical practice. Table 1 summarizes representative studies across diverse surgical indications, highlighting typical dose ranges, delivery vehicles, patient-specific variables, and key outcomes. This comparative overview underscores that rhBMP-2 efficacy is context-dependent and illustrates how adjusting dosage and scaffold selection according to the clinical scenario and patient characteristics can optimize bone regeneration while minimizing complications.

601-620

-

Recent preclinical and clinical data indicate that rationally engineered carriers can equal or surpass the performance of absorbable collagen sponges while attenuating adverse effects by localizing and smoothing rhBMP-2 release. Poly(ethylene glycol) “click” hydrogels achieved defect closure equivalent to collagen in murine calvaria yet produced less off-target mineralization, and tunable TG-PEG hydrogels similarly enhanced calvarial regeneration with stiffness-dependent delivery kinetics [91]). Keratin-based scaffolds have matched collagen for new-bone formation in a rat femoral defect while exhibiting ~4-fold higher local retention of fluorescently labeled rhBMP-2 and reduced distal biodistribution, consistent with improved containment [66]. Composite biphasic-ceramic/hydrogel putties that sustain rhBMP-2 release (e.g., hydroxyapatite/β-TCP/poloxamer formulations) have outperformed collagen sponges in a rat femoral nonunion model, yielding a 6-week union rate of 76.5% vs 35.3% and higher BV/TV and BMD [67]. Early clinical experience aligns with this trajectory: in a 2024 prospective multicenter TLIF study using E. coli–derived rhBMP-2 (0.5–1.0 mg per level with an HA putty), surgeons reported 100% CT-confirmed fusion at 52 and 104 weeks with no cases of seroma, radiculitis, cage migration, or ectopic bone [73]. Together, these findings support the premise that controlled-release matrices can maintain or enhance osteogenesis at a given rhBMP-2 dose, thereby reducing total protein burden and the likelihood of inflammatory or heterotopic responses while broadening the therapeutic window. 

744-754

In summary, while collagen carriers remain clinically established, ongoing research into advanced composites, biopolymeric matrices, and smart nanocarrier systems highlights the potential for safer, more effective, and patient-tailored delivery of rhBMP-2. At the same time, growing awareness of the limitations of conventional rhBMP-2 therapy has fueled interest in molecularly targeted alternatives capable of transient, dose-dependent osteogenic factor expression with minimal immunogenicity. A comparative summary of BMP-2 efficacy across various delivery carriers is provided in Table 2.

In summary, accumulated preclinical and clinical evidence indicates that the choice of delivery system is a decisive determinant of both efficacy and safety of rhBMP-2-based bone regeneration. Traditional collagen sponges, while historically instrumental for clinical translation, are characterized by rapid burst-release kinetics that frequently lead to radiographically detectable heterotopic ossification and early osteolytic changes, particularly when high or poorly contained doses are applied. Although these complications are often subclinical, their occurrence underscores the biological liability of unmodulated release. In contrast, ceramic and composite carriers that provide slower and more sustained release allow for lower total rhBMP-2 doses without loss of regenerative effect, achieve higher or equivalent fusion rates, and markedly reduce the risk of inflammatory reactions, ectopic ossification, and early bone resorption. A comparative summary of rhBMP-2 efficacy across various delivery carriers is provided in Table 3.

Comments 2: Furthermore, potential translational barriers such as regulatory approval, cost,

and scalability should be addressed to strengthen the practical impact of the review.

Response 2:

3) As recommended, we have added a dedicated subsection on translational barriers, addressing key issues such as regulatory approval pathways, economic constraints, and scalability of GMP-grade rhBMP-2 manufacturing.

Lines

It was originally in the text

Corrections

386 -477

-

5.1 Translational Barriers: Regulatory Approval, Cost, and Scalability

Despite promising efficacy, rhBMP-2 therapies face significant translational hurdles.  Regulatory agencies treat rhBMP-2 products as complex biologic-device combinations, requiring extensive safety and efficacy trials for each new formulation. Infuse Bone Graft, for instance, is FDA-approved only for specific indications, and any novel rhBMP-2 delivery system must undergo similarly rigorous approval. A realistic appraisal of translational barriers should note that rhBMP-2 approvals are indication- and device-specific, not blanket. In the United States, the FDA authorized InFUSE Bone Graft only in combination with the LT-CAGE for single-level ALIF, with subsequent supplements tied to specific implants rather than broad anatomic expansion; off-label use in the cervical spine was explicitly flagged in the FDA’s 2008 public health communication because of life-threatening airway edema, reinforcing the need for conservative dosing and strict localization in that region [18]. n the EU, InductOs (dibotermin alfa) likewise holds indication-specific marketing authorization, and its vulnerability to GMP disruptions was illustrated when the European Medicines Agency recommended suspension in 2015 due to quality defects at a component site; the European Commission lifted the suspension in 2017 only after GMP compliance was restore, showing how manufacturing robustness directly conditions clinical availability [62](ссылка)

Economic factors further constrain adoption. rhBMP-2 remains very costly: Infuse kits still run on the order of $2,500–$6,000 each [63], a price that has not decreased in over 20 years. High acquisition cost often outweighs perceived savings; some analyses report that autograft procedures (despite longer OR times) can match or beat total cost once all follow-up care is counted. For example, one study found that including revision surgeries made the net costs of rhBMP-2 versus autograft roughly equivalent. Large-scale manufacturing of rhBMP-2 under GMP also remains complex and expensive, limiting production scale [64]. A 2024 systematic review of cost-effectiveness analyses reported a baseline hospital acquisition cost for rhBMP-2 ranging from approximately $900 to $5,500 per case, varying by indication, dose, and institutional contracting. The review further noted that rhBMP-2 frequently entails higher upfront costs compared with autograft or demineralized bone matrix with local bone, for example $42,627 vs $38,686 at 1 year in a large lumbar fusion cohort and $97,917 vs $85,838 at 90 days in multilevel cervical fusion. These findings underscore that although rhBMP-2 can reduce revision surgery rates in selected high-risk patients, its routine use may impose a significant budget impact and is not uniformly cost-effective across all clinical scenarios [65]. Taken together, these data rationalize the common practice of reserving rhBMP-2 for scenarios in which a marginal gain in fusion probability materially alters outcomes - multilevel constructs, revisions, limited autograft, or compromised bone quality). Scalability hinges on complex, tightly regulated biomanufacturing. Your review already documents sharp differences in production yields across expression platforms (e.g., stable CHO ≈ 153 pg/mL vs HEK293 ≈ 280 ng/mL vs CHO cell-free ≈ 40 µg/mL in ~3 h), illustrating why consistent GMP supply remains costly and why combination-product kits remain expensive in practice; the EU suspension episode above underscores the real-world consequences of supply-chain fragility. Scalability likewise depends on complex, tightly regulated biomanufacturing. Large-scale GMP production of rhBMP-2 remains costly, and yields vary widely across expression platforms, for example, stable CHO cell lines typically produce only picograms per mL, whereas transiently transfected HEK293 cells reach nanogram levels, and modern cell-free systems can achieve tens of micrograms per mL within hours.

Given these considerations, rhBMP-2 use must be tailored to patient and clinical context. In spine fusion and fracture repair, rhBMP-2 typically offers only a modest absolute improvement in fusion rates over autograft. Therefore, most experts reserve rhBMP-2 for high-risk scenarios where enhanced healing is crucial (e.g. complex revisions, multi-level fusions, severe osteoporosis, or where autograft is insufficient). In routine cases (young, healthy patients or small defects), standard grafts are often preferred to avoid rhBMP-2’s costs and risks. Within rhBMP-2 use, the advent of new carriers allows dose and delivery to be customized. For example, a slow-release ceramic or hydrogel scaffold can be chosen for an elderly patient to enable a low BMP-2 dose with sustained effect, whereas a more aggressive rhBMP-2 regimen might be used for a large defect in a younger patient. Clinicians should apply the quantitative data to decision-making: knowing that a keratin matrix retains more rhBMP-2 locally [66] or that a putty carrier doubles fusion success in models can guide carrier choice. In all cases, the goal is to match the rhBMP-2 platform to the defect and patient factors – leveraging superior carriers and dosing protocols to maximize bone healing while minimizing ectopic bone and inflammation [67].

Comments 3: 1. The figure is original or requires copyrights?

Response 3:  We thank the reviewer for raising this important point. All schematic figures included in the manuscript were created by the authors specifically for this review and are original work. Therefore, no copyright permissions are required.

Comments 4: My piece of advice is to rephrase the Conclusions section in such a way that will reflect the

practical/clinical aspect of the current paper. It will increase the impact of the paper.

In conclusion, I appreciate very much the topic, the quality of the paper and the way in which the

paper is structured. Congratulations to the authors! Please receive my warmest regards and best wishes for future projects!

Response 4: We appreciate this valuable suggestion. We have revised the Conclusions section to place stronger emphasis on the practical implications of optimizing rhBMP-2 delivery strategies, including safety-oriented dosing, controlled-release carrier systems, and the translational potential of advanced molecular approaches such as cmRNA and regional gene therapy.

We believe that this change improves the clinical relevance and impact of the review’s final message.

Lines

It was originally in the text

Corrections

925-958

In light of the limited regenerative capacity of large bone defects, together with the clinical and ethical constraints of autologous transplantation, contemporary biomedical research is increasingly focused on developing effective and safe alternative strategies for skeletal repair. Among the most extensively investigated osteoinductive agents remains rhBMP-2, which possesses a strong capacity to stimulate bone formation. Yet, its wide-spread clinical use continues to be challenged by the need to optimize recombinant production methods, enhance protein stability, and precisely regulate its bioavailability. These challenges necessitate not only careful molecular refinement of the protein itself but also the rational design of delivery matrices capable of providing controlled and localized release. The expanding range of carriers available today, from traditional collagen scaffolds to sophisticated bioceramic–polymeric composites, has already made it possible to reduce the frequency of adverse effects and improve therapeutic outcomes by offering spatiotemporal control over rhBMP-2 release. In parallel, a new generation of molecularly oriented strategies has emerged, including cmRNA and regional gene therapy, both of which enable transient in vivo BMP-2 expression while fostering a pro-regenerative microenvironment. Notably, these approaches not only demonstrate strong osteogenic activity but also offer scalability and opportunities for personalization, thereby aligning with the broader goals of precision medicine. These platforms hold significant promise for the creation of clinically adaptable protocols that may eventually replace conventional reconstructive techniques. By integrating efficacy with safety and adaptability, they have the potential to meet the stringent demands of modern regen-erative medicine and transform the future of bone tissue engineering.

In light of the limited regenerative capacity of large bone defects, together with the clinical and ethical constraints of autologous transplantation, contemporary biomedical research is increasingly focused on developing effective and safe alternative strategies for skeletal repair. Among the most extensively investigated osteoinductive agents remains rhBMP-2, which possesses a strong capacity to stimulate bone formation. Yet, its wide-spread clinical use continues to be challenged by the need to optimize recombinant production methods, enhance protein stability, and precisely regulate its bioavailability.

These challenges necessitate not only careful molecular refinement of the protein itself but also the rational design of delivery matrices capable of providing controlled and localized release. The expanding range of carriers available today, from traditional collagen scaffolds to sophisticated bioceramic–polymeric composites, has already made it possible to reduce the frequency of adverse effects and improve therapeutic outcomes by offering spatiotemporal control over rhBMP-2 release.

In parallel, a new generation of molecularly oriented strategies has emerged, including cmRNA and regional gene therapy, both of which enable transient in vivo BMP2 expression while fostering a pro-regenerative microenvironment.

At the same time, it is increasingly clear that the clinical value of rhBMP-2 depends on careful, case-specific application. Across trials, rhBMP-2 typically improves fusion rates by only single-digit absolute percentages, meaning its use should be reserved for situations where that modest benefit outweighs the risks and costs. Emerging delivery technologies, such as hydrogels, nano-composites, and protein-based scaffolds, are beginning to tip this balance by achieving similar bone regeneration with lower doses and fewer complications. Still, high manufacturing costs, complex regulatory approval pathways, and scalability remain major barriers to broader clinical adoption. To move forward, future rhBMP-2–based therapies will need to balance potency with safety, demonstrate cost-effectiveness, and be matched to clearly defined clinical needs. By emphasizing evidence-based risk–benefit tradeoffs, patient-specific strategies, and translational feasibility, next-generation delivery systems can help make rhBMP-2 therapies both practical and clinically impactful.

4. Response to Comments on the Quality of English Language

Point 1:

Response 1:    (in red)

5. Additional clarifications

We are sincerely grateful to the reviewer for the constructive and thoughtful feedback. The revisions have, in our view, substantially strengthened the manuscript, enhancing its logical structure, mechanistic depth, and practical relevance. It is our hope that the changes satisfactorily address all concerns and that the revised manuscript will serve as a valuable reference for both researchers and clinicians.

We look forward to the editorial decision on the revised submission and remain available to address any further questions or comments that may arise. Our sincere thanks are also extended to the editors of the International Journal of Molecular Sciences for their careful consideration and support throughout the review process.

With kind regards,

The team of authors.

Round 2

Reviewer 1 Report

Comments and Suggestions for Authors

after successful revision this manuscript can be recommended for publication